# Comparison of the optical depth of total ozone and atmospheric aerosols in Poprad-Gánovce, Slovakia

Peter Hrabčák[1]

[1]Slovak Hydrometeorological Institute, Poprad-Gánovce, 058 01, Slovakia

*Correspondence to*: Peter Hrabčák (peter.hrabcak@shmu.sk)

**Abstract.** The amount of ultraviolet solar radiation reaching the Earth's surface is significantly affected by atmospheric ozone along with aerosols. The presented paper is focused on a comparison of the total ozone and atmospheric aerosol optical depth in the area of Poprad-Gánovce, which is situated at the altitude of 706 metres above sea level in the vicinity of the highest mountain in the Carpathian mountains. The direct solar ultraviolet radiation has been measured here continuously

since 1994 using a Brewer MK IV ozone spectrophotometer. These measurements have been used to calculate the total amount of atmospheric ozone and, subsequently, its optical depth as well. They have also been used to determine the atmospheric aerosol optical depth (AOD) using the Langley plot method. Results obtained by this method were verified by means of comparison with a method that is a part of the Operational Brewer Program, as well as with measurements made by a Cimel sunphotometer. Diffuse radiation, stray-light effect and polarization corrections were applied to calculate the AOD

using the Langley plot method. In this paper, two factors that substantially attenuate the flow of direct ultraviolet solar radiation to the Earth's surface are compared. The paper presents results for 23 years of measurements, namely from 1994 to 2016. Values of optical depth were determined for the wavelengths of 306.3 nm, 310 nm, 313.5 nm, 316.8 nm and 320 nm. A statistically significant decrease in the total optical depth of the atmosphere was observed with all examined wavelengths. Its root cause is the statistically significant drop in the total optical depth of aerosols.

**1 Introduction**

It is known that anthropogenic changes in the amount of total ozone and atmospheric aerosols have a considerable impact on the solar UV radiation reaching the Earth's surface (De Bock et al., 2014; Czerwińska et al., 2016). An increased transmittance of UV radiation through Earth's atmosphere has a manifest influence on human health and natural ecosystems. Higher doses of UV radiation have adverse effects mainly on terrestrial plants that are exposed to it on a long-term basis

(Jansen et al., 1998). Exposure of human organism to excessive UV radiation doses may cause premature ageing of the skin, weakening of immune system, and damage to cells and DNA, which may consequently lead even to skin cancer and other health conditions (Greinert et al., 2015). Positive effects of UV radiation are known as well, which include vitamin D production in the skin in particular. This vitamin is much needed for proper functioning of a human organism (Kimlin and Schallhorn, 2004). In the past, anthropogenic impact led to the increased transmittance of solar UV radiation through Earth's

atmosphere as a result of a decrease in the total amount of ozone. Global ozone layer depletion began to show more significantly in the 1980s, reaching its peak with the value of ca 5 % (with respect to the average for the period of 1964–1980) in early 1990s. The depletion has already been reduced in recent years, amounting to ca 3 % on average for the whole Earth (WMO, 2014). In the middle latitudes of the Northern Hemisphere (35° N–60° N), the depletion of ozone layer

reached 3.5 % around 2010 (2008–2012). The depletion of up to 6 % was peculiar to the middle latitudes of the Southern Hemisphere (35° S–60° S) in the same period (WMO, 2014).

On the other hand, anthropogenic emission of aerosols into the atmosphere causes reduction of solar UV radiation reaching the Earth's surface, especially in industrialized areas. In the early 1990s, it was determined that solar UV-B radiation had decreased by ca 5–18 % in non-urbanized areas of industrialized countries since the Industrial Revolution as a result of air

pollution (Liu et al., 1991). Anthropogenic aerosols may reduce the UV radiation reaching the Earth's surface by more than 50 % even in highly polluted urban areas (Krotkov et al., 1998; Sellitto et al., 2006). Anthropogenic emissions of aerosols have been gradually reduced in the developed countries, and a drop in the aerosol optical depth (AOD) has been observed in several locations (Kazadzis et al., 2007; Mishchenko and Geogdzhayev, 2007; Alpert et al., 2012; de Meij et al., 2012; Zerefos et al., 2012). Aerosols have a substantial effect on other physical and chemical processes taking place in the

atmosphere as well (Seinfeld and Pandis, 2006; Raghavendra Kumar et al., 2010). They affect predominantly the chemical composition of the troposphere and, in certain cases, of the stratosphere as well, primarily with regard to major solar eruptions or flights of aircraft (Finlayson and Pitts, 2000; Seinfeld and Pandis, 2006). They can reduce the visibility (Lyamani et al., 2010) and also have a significant effect on human health in many cases (WHO, 2006). The presence of aerosols in the atmosphere has an impact on the energy balance of the Earth as well, namely directly, semi-directly and

indirectly (De Bock et al., 2010).

The direct impact means scattering and absorption of shortwave and longwave radiation. Absorption of radiation subsequently leads to warming of those atmospheric parts, where aerosols are present (primarily in the boundary layer of the atmosphere) and higher temperature consequently leads to evaporation of cloud layers. The last sentence concisely described the semidirect impact, resulting in a higher density of solar radiation flow reaching the Earth's surface (Cazorla et al., 2009).

The higher temperature may lead to a change of thermal stratification of the atmosphere, which consequently affects vertical and horizontal movements of air in the atmosphere. The indirect impact pertains to the ability of aerosols to act as condensation nuclei or ice nuclei, which affects microphysical and optical properties of clouds. Yet, it concerns a change of their radiation characteristics, change of atmospheric precipitation characteristics, and alteration in cloud lifetime as well. An increase in the number of condensation nuclei leads to the rise of cloud droplet count and to the reduction of their size under

the given conditions of water content in the atmosphere, causing an increase of albedo and extension of cloud lifetime (Lohmann and Feichter, 2005; Unger et al., 2009). For the said reasons, the anthropogenically emitted aerosol particles considerably contribute to the ongoing global climate change, while their influence on radiation balance is still uncertain to a great extent (IPCC, 2014, and references therein).

This paper is focused primarily on the aerosol optical depth obtained by measurements made using the Brewer ozone spectrophotometer and presents one of the possible methodical approaches to its calculation. The presented method is verified by means of an alternative method of AOD calculation and is also verified by Cimel sunphotometer measurements. In the paper, obtained values of AOD are compared with the impact of total atmospheric ozone optical depth on the reduction of solar UV radiation. The Brewer spectrophotometer allows determining the optical depth in the UV region of the spectrum for the wavelengths of 306.3 nm, 310 nm, 313.5 nm, 316.8 nm and 320 nm. It was determined for the period of 1994–2016. The examined series is 23 years long, which enabled to quantify a linear trend of examined optical depth characteristics.

The employment of Brewer spectrophotometer measurements to determine the AOD as well has already been published in multiple studies. Some of them presented only short series of measurements. Therefore, they do not include multiannual trends of AOD (Carvalho and Henriques, 2000; Kirchhoff et al., 2001; Marenco et al., 2002). It is mainly more recent papers that present also multiannual measurements, but they do not always present trend information as well. It is stated in Jaroslawski et al. (2003) that the development of AOD for the Polish Belsk station in the period of 1992–2002 appears to have no trend. A quantification of the trend for the Belgian Uccle station can be found in Cheymol and De Backer (2003). It indicates a significant negative trend at the level of $2\sigma$ for all wavelengths for Brewer MKII spectrophotometer measurements in the period of 1989–2002. For example, the trend value for 320.1 nm is –2.13 ±0.39 %/year. It also states that equally significantly negative trend was not observed for the period of 1984–2002. The trend value for the above wavelength was only –0.78 ±0.42 %/year in that period. The trend is also mentioned in Kazadzis et al. (2007), namely for the Greek Thessaloniki station in the period of 1997–2005. Data analysis for the Brewer MKIII spectrophotometer adjusted for the seasonal cycle using a linear regression showed the trend of –2.9 ±0.92 %/year for the wavelength of 320.1 nm. It was also determined by a Student's t-test that the statistical significance of this change was more than 99 %.

## 2 Methodology

### 2.1 Location of experiment

The Brewer ozone spectrophotometer (MKIV) and the Cimel sunphotometer are located on the roof of a building of the Aerological and Radiation Centre, Slovak Hydrometeorological Institute (SHMI), in Gánovce near the city of Poprad. Their coordinates are 49.03°NW lat. and 20.32 EW long. with the altitude of 706 m above sea level. The aerosol content in the air, both total amount and composition by types, is determined by local sources on one hand, and by atmospheric circulation on the other hand, which can move a certain air mass together with aerosols even for several thousand kilometres. In isolated instances, it may also concern a transport of Sahara dust from Africa. The Sahara dust was present over the Slovak Republic at least for 20 days in 2016 (Hrabčák, 2016). Major local sources include products of solid fuel combustion in adjacent municipalities and the agriculture. A bare dry soil or even plant products are often blown away by wind, as the location is rather windy. The proximity of the city of Poprad (ca 1.5 km) with the population of ca 53,000 and various industrial

activities plays a certain role as well. On the other hand, it is a submontane location, since the highest mountain of the Carpathian mountains (2,655 metres above sea level) is situated only 20 km away from the station. In spite of the proximity of the mentioned city, the area can be deemed rural in general with respect to the anthropogenic impact.

## 2.2 Instrumentation

The Brewer ozone spectrophotometer (No. 97, a single monochromator – model MKIV) is a scientific instrument operating in the ultraviolet and visible region of the solar spectrum. The Brewer spectrophotometer was originally designed to measure the vertical column of ozone in the atmosphere (Brewer, 1973). More recent instruments (including the model MKIV) allow measuring the vertical column of $SO_2$, $NO_2$ (for this purpose, measurements of solar radiation are made in the visible region of the spectrum) and global UV radiation as well (from 290 to 325 nm, with an increment of 0.5 nm). The instrument breaks

down the solar radiation reaching the Earth's surface using its optical system and selects predetermined wavelengths with higher and lower absorption of $O_3$ and $SO_2$ from the ultraviolet part of its spectrum. On the basis of different radiation absorption for the selected wavelengths, it is possible to derive the total amount of given gases in the vertical column of the atmosphere (using the DS (direct sun) or ZS (zenith sky) measurements). The spectral separation of solar radiation is carried out by means of a modified Ebert f/6 spectrometer, which uses a holographic diffraction grating with a resolution of 1,200

lines per one mm (Sci-Tec, 1999). The instrument stability of Brewer spectrophotometer measurements is ±0.01 nm, namely throughout the temperature range (Sci-Tec, 1999). The value of the smallest wavelength increment (microstep) is 0.006 ±0.002 nm (Sci-Tec, 1999).

It performs standard measurements of direct solar radiation (DS) in the UV region of the Brewer spectrophotometer at five selected wavelengths, namely 306.3 nm, 310 nm, 313.5 nm, 316.8 nm and 320 nm. They are not completely identical for

individual instruments, which is given by minor differences in the position of a measurement slit and other minute mechanical and optical differences (Savastiouk and McElroy, 2005). The aforementioned five wavelengths represent their long-term average, as very small changes in their size occur over the years. During the monitored 23 years, the sizes of wavelengths were refined five times in total. Their accurate sizes were used at the beginning of the examined period. In 1994, the sizes were 306.276 nm, 310.04 nm, 313.494 nm, 316.799 nm and 319.999 nm. The end of the period, that is 2015,

was characterized by values of 306.276 nm, 310.035 nm, 313.476 nm, 316.755 nm and 319.989 nm. Observed deviations are only minimal. Therefore, the aforementioned long-term average is generally applicable to the whole period. The instrument No. 097 has undergone regular 2-year calibrations and daily tests using internal lamps (mercury and standard lamp) from the start of the measurements (18 August 1993). The calibrations are provided by International Ozone Services (IOS). The instrument is calibrated against the World Brewer Reference Triad (World Meteorological Organization standards),

maintained by Environment Canada, by means of a portable reference instrument No. 017.

Direct solar radiation measurements can be used to determine the aerosol optical depth as well. This optical property can be determined in the ultraviolet region of the solar spectrum for the five aforementioned wavelengths, for which the DS measurement is performed by default. It is known that the utilization of Brewer spectrophotometer to calculate the AOD

encompasses particular sources of potential systematic errors. If these errors are neglected, it may lead to negative values of the Ångström exponent for the examined wavelengths (Arola and Koskela, 2004). The first source of errors is the impact of undesired diffuse radiation transmittance on the DS measurements. The second source is the daily cycle of ozone in urbanized areas. The third source is the neglect of $NO_2$ absorption impact. In addition, the stray-light effect is the fourth source for the single monochromator. It is stated in Arola and Koskela (2004) that the neglect of diffuse radiation impact is potentially the biggest source of errors. On the contrary, the neglect of stray-light effect has the lowest impact. Unlike the AOD for the shortest and the longest wavelength, its impact in the cited paper is approximately 7-fold lower than the diffuse radiation impact. In the light of the listed errors, it is not recommended to determine extraterrestrial constants (ETCs) using the Langley plot method (LPM) for low altitude stations in urbanized areas, unless corrections for the impact of diffuse radiation and the daily cycle of ozone are known. The impact of daily ozone cycle and $NO_2$ can be neglected considering the rural location and higher altitude of the Poprad-Gánovce station. The diffuse radiation impact and stray-light effect were not neglected. A recommendation to ensure the air mass factor does not exceed the value of 3 in the calculation of ETCs was taken into account as well. It is important to mention that the polarization effect is another potential source of systematic errors as well (Cede et al., 2006). The polarization effect for the instrument No. 97 was not determined directly. It is assumed that the dependence of polarization effect on the zenith angle is similar for all types of Brewer spectrophotometers (Cede et al., 2006). As a result, corrections published in Cede et al. (2006) could be applied to the instrument No. 97. The impact of temperature changes was not also neglected. The corrections resulting from the changes in temperature of the instrument are allowed for in the adjustment of raw data.

The CE 318 NE dP automatic Cimel sunphotometer was used to verify the AOD values obtained by the Brewer spectrophotometer (Cimel – advanced monitoring, 2015). It is an instrument that enables to measure the direct, diffused and polarized solar radiation. It performs measurements of direct solar radiation for the selected wavelengths in the ultraviolet, visible and infrared region of the spectrum (Cimel – advanced monitoring, 2015). AOD values obtained for the wavelengths of 340 nm and 380 nm were used for comparison with the measurements of Brewer spectrophotometer. The sun photometer began to perform first measurements at the Poprad-Gánovce station on 12 December 2014. If the conditions allow, it is in automatic operation every day except for the calibration period. For that reason, no continuous measurements are available. The calibration period is approximately 2 months long, and the instrument is away from the station during that time. The calibration is provided within the AERONET (AErosol RObotic NETwork) global network and is carried out by Service National d'Observation PHOTONS/AERONET, Laboratoire d'Optique Atmosphérique, CNRS-Université de Lille. Only the level 2.0 data were used in this paper. It is the highest possible level defined within the AERONET.

## 2.3 Calculation of total ozone optical depth

Measured values for the total column ozone amount (TCO) obtained by the direct sun (DS) procedure were used to calculate the total ozone optical depth. A DS measurement is performed only with the relative optical mass of less than 4 and it takes approximately 2.5 minutes. During that time, the density of solar radiation flow is measured five times for each of the five

wavelengths. Thus five values of total ozone in Dobson units (DU) are obtained from a single DS measurement, which are consequently used to calculate an average and a standard deviation. Only the measurements that meet the standard deviation criterion (STDEV ≤ 2.5 DU) are selected for further data analysis. The total ozone was calculated using the Brewer Spectrophotometer B Data Files Analysis Program software v. 5.0 by Martin Stanek (http://www.o3soft.eu/o3brewer.html).

The optical depth was calculated for each accepted value of the total ozone. The calculation is represented by the following equation:

$$\tau_{\lambda,O_3} = \Omega_{O_3}\alpha(\lambda, T) = \Omega_{O_3}\sigma(\lambda, T)n \,, \tag{1}$$

where $\tau_{\lambda,O_3}$ is the total ozone optical depth, $\Omega_{O_3}$ is the total ozone in Dobson units, $\alpha(\lambda, T)$ is the absorption coefficient for ozone, $\sigma(\lambda, T)$ is the effective absorption cross-section of the ozone molecule (it is usually quantified for 1 cm²) and $n$ is the molecule count in the volume determined by 1 DU and 1 cm²; for $O_3$, it is a constant with the value of $n = 2.687 * 10^{16}$ (Schwartz and Warneck, 1995). It is recommended in Carlund et al. (2017) to utilize the same effective absorption cross-sections of the ozone molecule to calculate both the TCO and its optical depth, which is required to determine the AOD. The Operational Brewer Program for the Brewer spectrophotometer utilizes the effective absorption cross-sections of the ozone molecule, which are determined on the basis of Bass and Paur measurements (Bass and Paur, 1985). This scale is employed to calculate the total amount of ozone within the network of Brewer spectrophotometers according to the recommendations of the International Ozone Commission (http//www.esrl.noaa.gov/gmd/ozwv/dobson/papers/coeffs.html).

Today, more recent and accurate values of effective absorption cross-sections are already available based on measurements of the Molecular Spectroscopy Lab, Institute of Environmental Physics (IUP), University of Bremen, (http://www.iup.physik.uni-bremen.de/gruppen/molspec/databases/index.html; Gorshelev et al., 2014; Serdyuchenko et al., 2014). More recent values according to the IUP were used to calculate the ozone optical depth and the AOD. In order to preserve the consistency between the calculation of optical depth and the calculation of the TCO, it was necessary to calculate the TCO values according to a more recent set of IUP. This calculation was carried out as per the recommendations in Redondas et al. (2014). The dependence of effective cross-section on the temperature is very important. A so-called effective temperature for the given gas is usually used. In case of ozone measurements using the Brewer spectrophotometer, an average standard effective temperature of –45 °C (228.15 K) is defined (Redondas et al., 2014). Another difference in the calculation of the TCO compared to the Operational Brewer Program pertained to the used absorption coefficients for Rayleigh scattering. Instead of coefficients used by default, the value of TCO was determined with coefficients according to Bodhaine et al. (1999), which is in line with coefficients used to calculate the optical depth.

**2.4 Calculation of aerosol optical depth**

The Langley plot method (LPM) was employed to calculate the AOD. It is a traditional method employed to calculate the AOD by the Brewer spectrophotometer (Carvalho and Henriques, 2000; Kirchhoff et al., 2001; Silva and Kirchhoff 2004;

Cheymol et al., 2006; Sellitto et al. (2006). This method requires stable atmospheric conditions in order to determine the extraterrestrial constant (ETC). It needs mainly a low variability of the total ozone and atmospheric aerosols during the day, for which the ETC is determined. It is also necessary to avoid an impact of cloudiness on direct solar radiation (DS) measurements and to ensure a sufficient scope of zenith angles of individual DS measurements during the day, which is needed for the given method.

For the said reasons, this method is most appropriate for lower latitudes (especially in montane regions near the tropics), and it has certain limitations in the middle and particularly higher latitudes (Nieke et al. 1999; Marenco 2007). This method of AOD calculation has already been applied to the examined location of Poprad-Gánovce in the past (Pribullová, 2002). An alternative method of AOD calculation, developed by Vladimir Savastiouk, has been available since 2005 (Savastiouk and McElroy, 2005; Savastiouk, 2006; Kumharn et al., 2012). In this case, an algorithm of AOD calculation is a part of the Operational Brewer Program. Hence, the given method will be referred to as the Brewer software method (BSM) in this paper. The fundamental difference against the previous method is that ETCs for individual wavelengths are not determined using the LPM. They are obtained during calibration of the instrument, hence once in 2 years. Their size is determined during calibration based on a comparison with the portable reference instrument No. 017. Neglected changes in the sensitivity of the instrument over shorter time periods represent a disadvantage of both methods, as the ETCs are fixed for a longer period – 2 years in both cases. The LPM applied in this paper employs fixed ETCs for a 2-year intercalibration period, which is identical with the standard intercalibration period for the measurement of ozone. It is assumed that any significant service modifications to the Brewer spectrophotometer during calibration may affect both the calculation of ozone and the calculation of AOD. For that reason, the period not exceeding 2 years was used.

To calculate the AOD, it was necessary to apply the Beer–Bouguer–Lambert law:

$$S_\lambda = S_{0,\lambda}\, e^{-\mu_w\,\tau_\lambda} = S_{0,\lambda}\, e^{-\mu_{O_3}\,\tau_{\lambda,O_3} - \mu_r\,\tau_{\lambda,r} - \mu_a\,\tau_{\lambda,a}} = S_{0,\lambda}\, e^{-\mu_{O_3}\,\alpha(\lambda,T)\,\Omega_{O_3} - \mu_r\,\frac{\beta(\lambda)\,P}{P_{std}} - \mu_a\,\tau_{\lambda,a}} \quad , \tag{2}$$

where $S_\lambda$ is the flux density of solar radiation flow for the selected wavelength expressed by photon count per unit of time on Earth's surface, $S_{0,\lambda}$ is the flux density of solar radiation flow for the selected wavelength expressed by photon count per unit of time above Earth's atmosphere (extraterrestrial constant – ETC), $\tau_\lambda$ is the total optical depth of atmosphere, $\tau_{\lambda,O_3}$ is the optical depth for ozone, $\tau_{\lambda,r}$ is the optical depth for Rayleigh scattering, $\tau_{\lambda,a}$ is the optical depth for aerosols, and $\mu_w$ is the air mass factor for atmosphere as a whole. Its value was calculated as a weighted arithmetic average of individual aforementioned components, while the optical depth of a given component was the weighting factor.

Furthermore, $\mu_{O_3}$ is the airmass factor of the ozone layer determined according to Komhyr (1980), $\mu_r$ is the air mass factor for Rayleigh scattering determined according to Kasten and Young (1989), $\mu_a$ is the air mass factor of aerosols determined according to Kasten (1966), assuming that $\mu_a \cong \mu_{H_2O}$, $\alpha(\lambda,T)$ is the absorption coefficient for ozone, $\Omega_{O_3}$ is the total amount of ozone in Dobson units, $\beta(\lambda)$ is the normalized optical depth for Rayleigh scattering (for the standard atmospheric air pressure and the vertical column), $P$ is the atmospheric air pressure in the location of observation (the daily average was

used), and $P_{std}$ is the standard atmospheric air pressure (101,325 Pa). The contribution of sulfur dioxide was neglected, namely due to its low impact and due to its inaccurate determination as well.

The normalized optical depth for Rayleigh scattering $\beta(\lambda)$ was calculated according to Bodhaine et al. (1999). The use of coefficients according to Bodhaine et al. (1999) is in compliance with the recommendations

of the NOAA (National Oceanic and Atmospheric Administration). An NOAA document (https://www.esrl.noaa.gov/gmd/grad/neubrew/docs/RayleighInBrewer.pdf) states that standard coefficients for the Operational Brewer Program must not be used to calculate the AOD. It is stated in Carlund et al. (2017) that the total amount of ozone calculated using the standard coefficients is higher compared to the use of coefficients according to Bodhaine et al. (1999).

The value of $S_\lambda$ was obtained by adjustment of raw data (raw counts). It is essential to keep the sequence of the following steps in their adjustment. In the first step, raw counts saved in a B-file were converted to count rates. In the second step, deadtime compensation was applied. After the deadtime compensation, a correction was applied to the stray-light effect. In the fourth step, a correction for the temperature dependence was applied, including a correction for the utilized neutral density (ND) filter. These filters are automatically selected by the Brewer spectrophotometer with respect to the current

density of the solar radiation flow. There are 5 ND filters and 5 wavelengths as well, so 25 attenuation values are needed in total. The attenuation values of given filters are determined during calibration of the instrument. The impact of polarization was compensated in the fifth step. The last sixth step involved the correction due to the impact of diffuse radiation on DS measurements. When the above criteria are applied, five initial values of $S_\lambda$ will be obtained from a single DS measurement, and subsequently, the AOD will be calculated from each of them. The final AOD for the given

DS measurement is calculated as an arithmetic average of five values.

The ETC $S_{0,\lambda}$ was determined using the Langley plot method. The Langley plot method employs multiple measurements of the direct solar radiation at various zenith angles of the Sun in the sky. Its fundamental principle is as follows: The above Eq. (2) is adjusted by applying a natural logarithm:

$$\ln(S_\lambda) = \ln(S_{0,\lambda}) - \mu_w \tau_\lambda . \tag{3}$$

There is one equation available for every single measurement of the direct solar radiation, while $\mu_w$ and $\ln(S_\lambda)$ are the knowns, and $\ln(S_{0,\lambda})$ and $\tau_\lambda$ are the unknowns. A number of equations equal to the number of measurements will be acquired by applying multiple measurements at various zenith angles of the Sun. Theoretically, it is feasible to determine the unknowns already from two measurements, but for the practical purposes, it is advisable to acquire as many measurements as possible. That will guarantee a higher accuracy of the result. It is essential to linearly interpolate the obtained dependence

of natural logarithm of the solar radiation flow density $\ln(S_\lambda)$ on the total air mass factor of the atmosphere $\mu_w$ using the method of least squares. The inclination of obtained line $a$ (from the equation of a straight line $y = ax + b$) equals $\tau_\lambda$. The natural logarithm of ETC $\ln(S_{0,\lambda})$ is obtained when $x$ (in the equation of a straight line) equals 0 ($x$ represents $\mu_w$). The ETC for the given wavelength is valid for the entire intercalibration period, hence for 2 years. Its determination follows

the following procedure: The ETCs for individual wavelengths can be determined solely on days that meet the following conditions:

1. The air mass factor for the atmosphere as a whole is less than 3.

2. The AOD calculated as an average of five values within a single DS measurement for the wavelength of 320 nm is less than 0.5.

3. The difference between the maximum and minimum value of AOD within a single DS measurement is less than 0.03.

4. The number of direct solar radiation measurements is at least 50 (i.e., 10 DS measurements).

5. The difference between the maximum and minimum air mass factor for the atmosphere as a whole is greater than 1.

6. The standard deviation from the measured values of the total ozone on the given day is less than 2.5.

7. The standard deviation from the measured AOD values on the given day is less than 0.07.

8. $\left|\ln(S_\lambda)_i - \ln(S_\lambda)_j\right| < 1.75 * \frac{\sum_{i,j=1}^{n} \left|\ln(S_\lambda)_i - \ln(S_\lambda)_j\right|}{n}$, where $\ln(S_\lambda)_i$ is the value of $\ln(S_\lambda)$ obtained from the equation of linear interpolation after the substitution of a specific $\mu_w$, $\ln(S_\lambda)_j$ is the actually measured value of $\ln(S_\lambda)$ at the given value of $\mu_w$, and $n$ is the total number of direct solar radiation measurements on the given day. The fundamental principle is to exclude measurements that have too high values of residues of the linear interpolation (greater than or equal to 1.75 times the average of the residues on the given day). The threshold value of 1.75 in the above equation was selected based on the results of an optimization test. The goal of the test was to acquire the maximum possible number of days with very good linear interpolation (the selected days had to meet the ninth condition).

9. The determination coefficient for the linear interpolation is greater than 0.98.

The conditions defined above were applied respectively. The following criterion was applied to all determined ETCs within the given intercalibration period:

$$\frac{|ETC - AVERAGE(ETCs)|}{STDEV(ETCs)} < 1.5 ,$$
(4)

where AVERAGE (ETCs) is an average of the determined ETCs and STDEV (ETCs) is a standard deviation. An average is calculated from the ETCs that meet the said criterion. This average is valid for the entire intercalibration period. The final values of ETCs used to calculate the AOD were available only after two iterations. At the beginning, it was not possible to apply the second and third criterion. It was possible to do so on the first and second iteration. The ETC varies around its mean value with regard to the distance of Earth from the Sun. Its correction was carried out according to the recommendation of the Guide to Meteorological Instruments and Methods of Observation (WMO, 2008). The calculation of apparent elevation and apparent zenith angle of the Sun in the sky was also made as per the recommendations of the WMO (WMO, 2008). The apparent elevation of the Sun constitutes the fundamental principle in the calculation of air mass factor.

All final values of AOD have undergone a cloud screening process, which is illustrated schematically in Fig. 1. Prior to the cloud screening process, all negative values of AOD were deleted. The first step of cloud screening is to delete the

DS measurements with a standard deviation for $O_3 > 2.5$ DU (described in more detail in Sect. 2.3). In the second step, all AOD values $\geq 1.5$ were deleted. Comparison of AOD values above 1.5 obtained from the Brewer spectrophotometer for the Belgian Uccle station determined that they could not be paired with Cimel sunphotometer measurements at all (De Bock et al., 2010). It proves that the limit of 1.5 is substantiated for Poprad-Gánovce station. The third step was to delete the

DS measurements with a difference between the maximum and minimum AOD value greater or equal to 0.03. The international Cimel sunphotometer network AERONET uses a limit for measurement triplet (three measurements per minute) with the value of 0.02 (https://aeronet.gsfc.nasa.gov/new_web/Documents/Cloud_scr.pdf). We decided to apply a less stringent criterion because our case involved five measurements performed in ca 2.5 minutes. The attached diagram shows that the limit of 0.03 has caused a considerable reduction in the total number of DS measurements. Application of the

second and third criterion has resulted in a reduction of the total number of measurements by up to 43 %. The DS measurements that satisfied all three previous criteria were included in a so-called daily database. A daily standard deviation of AOD (SDAOD) and a daily average of AOD (AAOD) were determined from these measurements. When SDAOD $< 0.015$, all AOD values in the daily database were transferred to a final database (Good AOD). When SDAOD $\geq 0.015$ in the fourth step, all DS measurements on the given day underwent the last fifth step. The following

applied to the fifth step: When the difference between the AOD for a given DS measurement and the daily average AAOD was $\geq 0.5$, the DS measurement was not included in the final database. A total of 36,497 DS measurements were included in the final database. That represents 57 % of the total number of measurements that entered the second step of cloud screening.

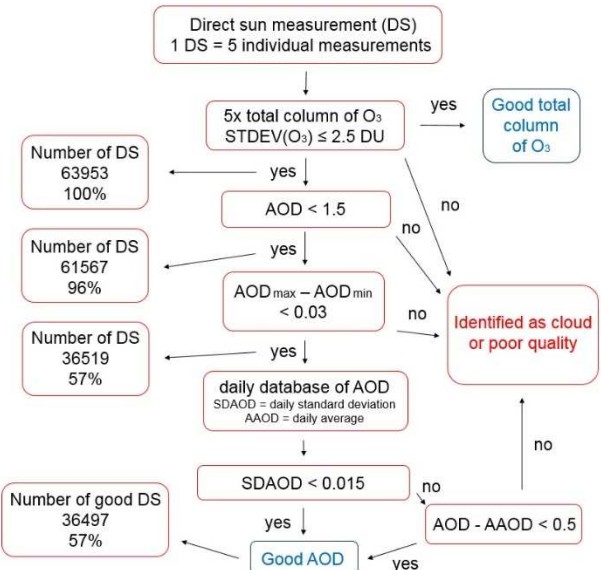

**Figure 1: Schematic illustration of cloud screening. The number of DS measurements that satisfy the respective criterion is**
**indicated on the left side.**

Comparison of AOD values obtained by the Cimel sunphotometer with AOD values from the Brewer spectrophotometer was not executed directly. First, the value of Ångström exponent was determined using pairs of wavelengths and their corresponding AOD values:

$$\alpha = -\frac{\log(\tau_{\lambda_1}/\tau_{\lambda_2})}{\log(\lambda_1/\lambda_2)}, \tag{5}$$

where $\lambda_1 = 340$ nm and $\lambda_2 = 380$ nm. On the basis of Ångström exponent and the known value of AOD (for the shorter wavelength), an AOD value for the wavelength of 320 nm was determined in the next step. This calculation was carried out for all individual measurements. The main goal of this calculation was to acquire a comparison of AOD for the same wavelength, which is more relevant.

The following procedure was employed to calculate individual characteristics of the total ozone and AOD optical depth: Daily averages were calculated as an arithmetic average of all values for the given day (minimum of one value). Monthly averages were calculated as an arithmetic average of such days, for which an average daily value was available. Annual averages were calculated as an arithmetic average of individual monthly values. A linear trend was calculated by means of a linear regression using the least squares method. An autocorrelation was not confirmed. Therefore, it was possible to determine the linear trend. The uncertainty of the linear trend is defined by the standard deviation (±σ) of the slope of the obtained linear dependence. The value of linear trend and the standard deviation was determined for a period of 10 years. The seasonal cycle was eliminated by annual averages.

## 3 Results and discussion

### 3.1 Correction for the diffuse radiation, stray-light effect and polarization

The correction of diffuse radiation impact on DS measurements was made pursuant to the recommendations in Arola and Koskela (2004). The fact that the full field of view reached the value of 2.6 for the Brewer spectrophotometer was taken into consideration. A ratio of the circumsolar radiation to the direct solar radiation was calculated using the SMARTS 2.9.5 programme (available at https://www.nrel.gov/rredc/smarts/). The calculations in SMARTS were implemented for rural aerosol conditions, which are characterized by the Ångström exponent equal to 0.96. The ratio of the circumsolar radiation to the direct solar radiation was determined for all five wavelengths. The values of zenith angle, aerosol optical depth and atmospheric air pressure were taken into account in the calculation of the correction factor. It follows that this factor was determined for specific conditions at a given time. In Fig. 2, there is a demonstration of correction factor values for five wavelengths and the selected sequence of zenith angles of the Sun. This demonstration characterizes the conditions that are close to the ones that may normally occur at the examined station. It can be seen that the value of the factor is primarily dependent on the zenith angle. It is also true that the value of the factor is inversely proportional to the size of a wavelength. The factor for the longest wavelength of 320 nm is by ca 7 % lower than the factor for the shortest wavelength of 306.3 nm.

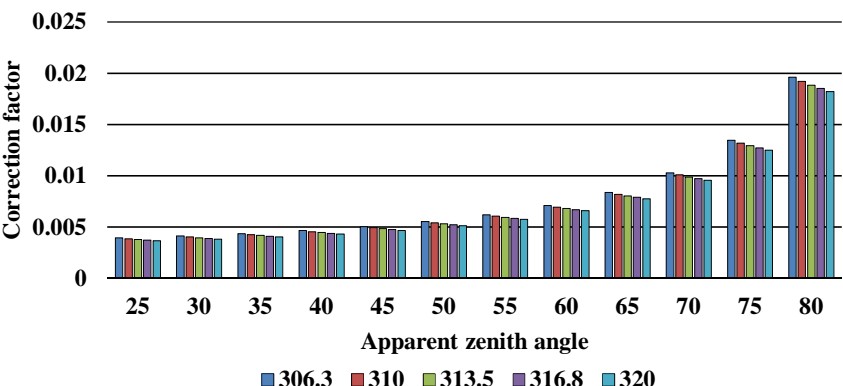

**Figure 2: Value of correction factor for the elimination of diffuse radiation impact for the selected zenith angles of the Sun, for five wavelengths for the conditions of AOD value of 0.3 (for 320 nm) and air pressure of 930 hPa.**

Detection of photons by the Brewer spectrophotometer is affected by noise photons received outside the analyzed wavelength. The noise photons are caused by radiation that does not follow the required optical path inside the monochromator due to the scattering on the grating, mirror, or housing. This problem is called the stray-light effect and is negligible for the double monochromators. The Brewer spectrophotometer at the Poprad-Gánovce station is classified as a single monochromator. For that reason, a correction for the stray-light effect was necessary (Arola and Koskela, 2004;

Garane et al., 2006). The correction was determined by analysing the spectral global UV radiation measured on cloudless days. It was feasible because the optical light path is the same for both the UV and DS measurement in the monochromator. In general, it is assumed that at the lowest wavelengths, i.e., below 292 nm, there is no transmittance of radiation to the Earth's surface due to the absorption in the atmosphere, and any signal measured must, therefore, be stray light. It is also assumed that the value of stray light is constant for all wavelengths.

Hence, a ratio of average count rates for four wavelengths in the region from 290 to 291.5 nm to the count rates for the monitored wavelength (one out of five) was determined for 3,386 spectral analyses in total, as well as for various zenith angles of the Sun within them. The value of determined correction factor is a function of the zenith angle. This dependence was described for the monitored wavelengths by a polynomial of the fourth degree. A demonstration is presented in Fig. 3. As a general rule, lower wavelengths are characterized by higher values of the correction factor. The observed difference

between the wavelength of 306.3 nm and 320 nm was approximately 10-fold. Furthermore, it was determined that the dependence on the zenith angle was not directly proportional for all monitored wavelengths. The direct proportion was seen only with the three shortest wavelengths.

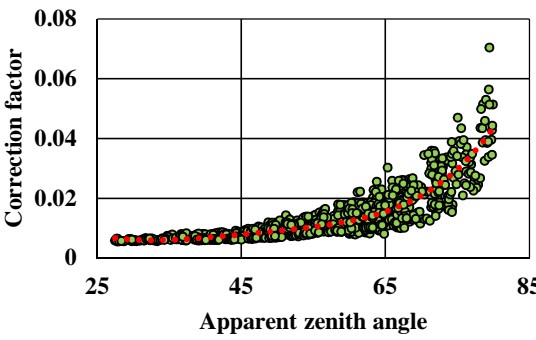 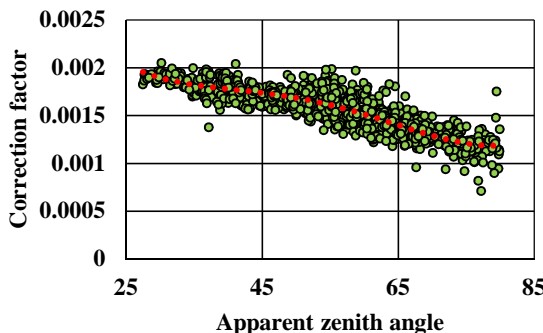

**Figure 3: Size of the correction factor for the elimination of stray-light effect depending on the apparent zenith angle for the wavelength of 306.3 nm (on the left) and for the wavelength of 320 nm (on the right).**

The impact of individual correction factors was tested on a shorter period, namely 2014. When the stray-light effect was allowed for, a drop in the difference in the AOD for the wavelengths of 306.3 nm and 320 nm was seen. Before the correction, the difference reached the value of –0.029, and after the correction, it was decreased by ca 0.006. This value is similar to the results in Arola and Koskela (2004). A correction eliminating the polarization of radiation followed after the correction for the stray-light effect. A drop in the difference with the value of 0.008 was observed again. It was then

followed by a correction for the impact of diffuse radiation. A drop in the difference was observed again, but it was lower, and its value was only 0.0004. This result is significantly lower compared to the result presented in Arola and Koskela (2004). In conclusion, it can be stated that application of all three corrections resulted in the drop in the difference in AOD for the shortest and the longest wavelength by 0.015 to the final value of –0.014. The negative value of Ångström exponent for the given pair of wavelengths was reduced, but it still persists.

**3.2 Extraterrestrial constants**

Strict conditions for the selection of appropriate days for the determination of ETCs and other mentioned criteria significantly eliminate obtaining of non-representative values. To calculate the ETC characterizing the entire 2-year period, 17 values of individual ETCs were employed with respect to the long-term average. This number is the same in case of all wavelengths. If the conditions were less strict, there would have been more days, for which it was possible to determine the

ETC. On the other hand, the spread of determined ETCs would be wider, which would have a negative effect on the required accuracy. Therefore, the chosen criteria represent an optimum compromise. The inaccuracy of ETC determination has a direct impact on the final values of AOD. On one hand, the root cause of such inaccuracy is weather influences, which considerably eliminate the number of days that are suitable for the determination of ETC. On the other hand, additional factors may include instrument instability.

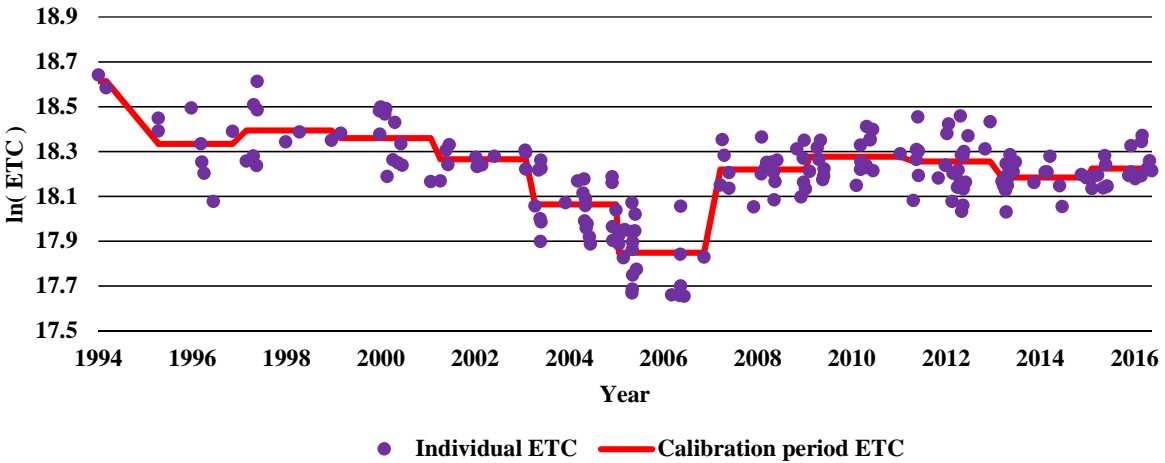

**Figure 4: Trend of ETCs values for the wavelength of 320 nm during 12 intercalibration periods, 1994–2016.**

The trend of ETCs (their natural logarithm) for the wavelength of 320 nm is illustrated in Fig. 4. It shows ETC values that
characterize the entire 2-year intercalibration period, as well ETCs determined on individual days in the given period.
The ETC for the monitored wavelength did not remain constant during the monitored period of 23 years. Particular changes
occurred in case of all five examined wavelengths. A total of 12 ETC values were determined for every single wavelength,
for each intercalibration period. The highest instability was observed in case of the longest wavelength. The value of its
variation coefficient is 18.2 %. By contrast, the lowest instability was observed for the wavelength of 313.5 nm. The value of
its variation coefficient is 15.2 %. In Fig. 4, you can notice that a more significant change in the value of ETC for the longest
wavelength occurred four times in total. The first change occurred between the first and second intercalibration period. There
is no specific explanation for this change. It is presumably related to the instability of ETCs during the first intercalibration
period. A shorter intercalibration period might have also caused the mentioned change. It resulted in lower number of days
that were suitable for the determination of ETC. Additional changes occurred between the fifth, sixth, seventh and eighth
intercalibration period. They were caused by issues with the instrument. For that reason, a secondary power supply board
had to be replaced in January 2005. In February 2007, a micrometer was replaced, and during calibration in May 2007, an
optical filter No. 3 was replaced and a BM-E80 high-frequency source was repaired as well.

Figure 5 presents a comparison of ETC values for individual wavelengths for the LPM and BSM in the last two
intercalibration periods. It can be seen that the values are similar. The first period is characterized by the fact that ETCs for
all five wavelengths for the LPM exceed the values for the BSM. It is exactly the reverse in the second period. In case of
both methods, there was a decrease in ETCs between the given two periods. The decrease for the BSM was more marked.

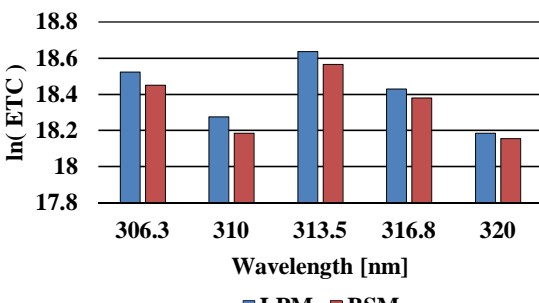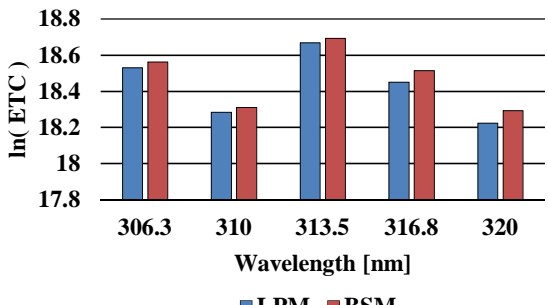

**Figure 5: Comparison of ETC values for individual wavelengths for the LPM and BSM in the period from 25 May 2013 to 19 May 2015 (on the left) and in the period from 20 May 2015 to 31 December 2016 (on the right).**

### 3.3 Comparison of AOD values obtained by LPM, BSM and the Cimel sunphotometer

The plausibility of results for the presented methodology of AOD calculation by means of the Langley plot method (LPM) was validated in two ways. In the first place, it was confronted with the results acquired by means of the Brewer software method (BSM). The given comparison is illustrated in Fig. 6. As has already been mentioned above, the BSM has been available since 2005. Its output has been plausible for the Poprad-Gánovce station only since calibration in 2013. Figure 6 consists of two graphs. The graph on the left side represents a comparison of AOD for the wavelength of 320 nm for the intercalibration period from 25 May 2013 to 19 May 2015. The graph on the right side covers the comparison of results for the same wavelength in the next intercalibration period from 20 May 2015 to the end of 2016. This division is necessary to point out the impact of used ETCs on the obtained results. In both intercalibration periods, only the measurements that had undergone the cloud screening process were selected. The mutual difference of AOD at the absolute value did not exceed the value of 0.1 (so-called large random error) in any of monitored periods. The total number of DS measurements in both intercalibration periods was 5,789. A total of 3,246 DS measurements were compared in the first period and 2,543 in the second period. An excellent agreement of both methods was seen in the first period. On one hand, it is proven by very high values of determination coefficient and correlation coefficient. On the other hand, the high level of agreement is proven by an average difference in AOD values for the LPM and BSM that reached –0.004. The following intercalibration period is also characterized by very high values of determination coefficient and correlation coefficient, yet they are somewhat lower than in the first period. A more significant disagreement was seen in case of mutual difference in AOD between the LPM and BSM that reached –0.054. The said differences between the LPM and BSM are not constant within the intercalibration period, as certain variability has been observed for them. The biggest differences within a year are observed in summer months.

Results for the remaining four wavelengths were compared in a similar fashion as well. For the wavelengths of 310 nm, 313.5 nm, 316.8 nm and 320 nm, the value of correlation coefficient reached at least 0.99. This value was not reached only

in case of the lowest wavelength of 306.3 nm in the first intercalibration period, where it was at the level of 0.97. The average of mutual differences in AOD between the LPM and BSM reached the absolute value of maximum 0.054 in the already mentioned case for the wavelength of 320 nm. For the lower wavelengths, the average of differences in the first period was always positive (0.031 on average). On the contrary, it was always negative in the second period (–0.028 on

average). Comparison of the AOD value and wavelength shows no unambiguous dependence for neither of the two methods. It applies to the LPM that the highest AOD was reached for the wavelength of 316.8 nm and the lowest AOD was reached for the wavelength of 320 nm with regard to the long-term average in the monitored period. It applies to the BSM that the highest AOD was reached for the lowest wavelength of 306.3 nm and the lowest AOD was reached for the wavelength of 310 nm with regard to the long-term average. Pribullová (2002) does not mention the unambiguous dependence of AOD on

the wavelength as well. It indicates the lowest AOD in case of the lowest wavelength and presents the highest values of AOD for the wavelength of 310 nm.

It follows that the LPM and BSM correlate nicely. On the other hand, random differences constitute a problem. They result in AOD values for the LPM that are lower or higher compared to the AOD values for the BSM. There are several causes of differences. It is not known that the BSM would take into account a change in the distance of Earth from the Sun.

The presented LPM takes the given change into account. Another distinction between the two methods is the fact that the BSM takes into consideration the total measured amount of $SO_2$ in the calculation of AOD. The presented LPM neglects the impact of $SO_2$. It may seem at a glance that the BSM is thus advantaged. However, the reverse is true. The Brewer spectrophotometer at the Poprad-Gánovce station fails to measure the total amount of $SO_2$ accurately. Even negative values are seen often. Another drawback of the BSM is the fact that it does not take into consideration corrections for the diffuse

radiation, polarization and stray-light effect. Values of determined ETCs affect the observed differences probably the most. These constants are fixed for both methods during the entire 2-year intercalibration period. An advantage of the LPM is that it takes into account the change of ETCs during the intercalibration period. The BSM utilizes ETCs determined during calibration. They are valid for 2 years and a potential change during the intercalibration period is not taken into account.

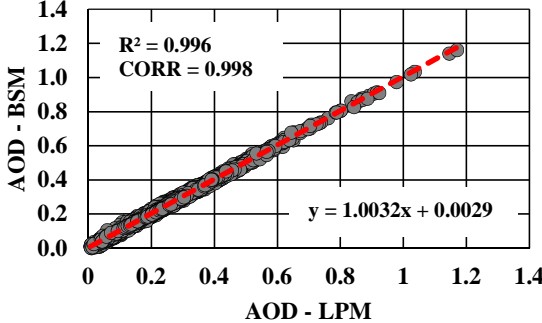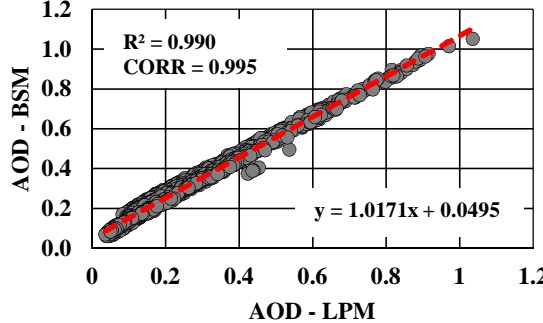

**Figure 6: Comparison of AOD values for the wavelength of 320 nm obtained by means of the LPM and BSM in the period from 25 May 2013 to 19 May 2015 (on the left) and in the period from 20 May 2015 to 31 December 2016 (on the right).**

Both presented methods were further compared with Cimel sunphotometer (CSP) measurements. These comparisons are illustrated in Fig. 7 (for the LPM) and Fig. 8 (for the BSM). The figures consist of two graphs just like in the previous case. In both intercalibration periods, only the DS measurements that had undergone the cloud screening process were selected. The comparison was made as follows: A CSP measurement was selected for an individual DS measurement from the Brewer spectrophotometer with the maximum allowed difference of 5 minutes between the two measurements. If there were several suitable measurements, the nearest one was selected. In case of comparison between the LPM and the CSP, there was no difference in AOD with the absolute value greater than 0.1. On the contrary, in case of comparison between the BSM and the CSP, there were as much as 222 instances of such DS measurements in total. A total of 199 measurements were compared in the first intercalibration period and 1,116 in the second period. It is apparent from the results of the comparison that both methods correlate very well with the CSP measurements. Yet the LPM shows higher values of the correlation coefficient in both intercalibration periods than the BSM. It was determined in the calculation of mutual differences that both methods matched the CSP measurements very well in the first intercalibration period. The difference between the LPM and the CSP was –0.02, and the difference between the BSM and the CSP was –0.01. In the second monitored period, the average difference between the LPM and the CSP reached the value of 0.02, and the average difference between the BSM and the CSP reached the value of 0.08. These average differences are the primary reason for observed offsets in attached graphs. As a result of the offset, the intersection of the linear fit is not the same as the intersection of the main axes of the graph. It is illustrated the best by the graph on the right in Fig. 8, because in this case, the average difference has the highest absolute value of all presented comparisons. In the light of the results, it can be stated that the LPM is the more reliable of the two methods.

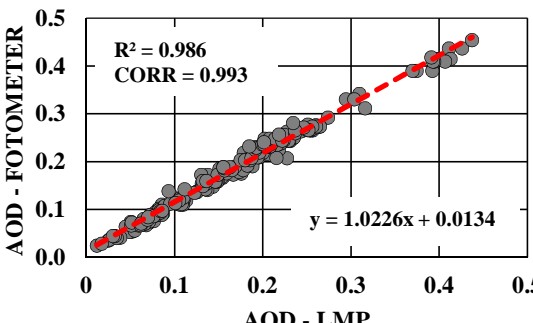 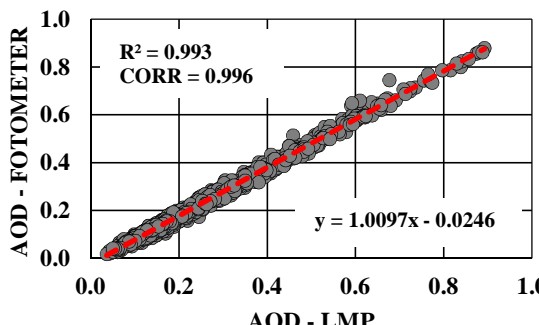

**Figure 7: Comparison of AOD values for the wavelength of 320 nm obtained by means of the LPM and the Cimel sunphotometer (CSP) in the period from 12 December 2014 to 19 May 2015 (on the left) and in the period from 20 May 2015 to 31 December 2016 (on the right).**

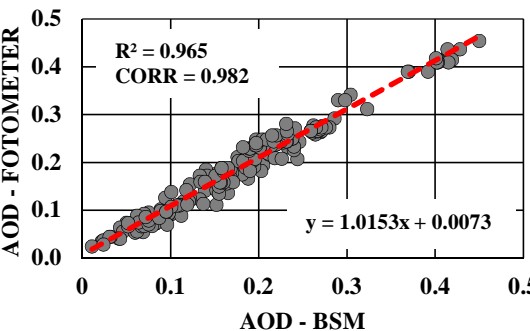 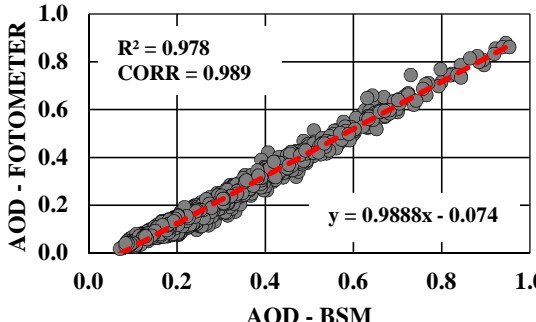

**Figure 8: Comparison of AOD values for the wavelength of 320 nm obtained by means of the BSM and the Cimel sunphotometer (CSP) in the period from 12 December 2014 to 19 May 2015 (on the left) and in the period from 20 May 2015 to 31 December 2016 (on the right).**

**3.4 Total ozone optical depth and AOD**

Measurements of total ozone from the Brewer spectrophotometer for the Poprad-Gánovce station are available from 1994. Over the last five years (2012–2016), the average value was 330 DU, which is 5 DU more than in the first five years of the monitored period (1994–1998). The linear trend for the period of 1994–2016 has the value of 2.6 ±2.3 DU for 10 years. If only DS measurements are taken into consideration, the value of the trend will be 0.8 ±2.2 DU for 10 years. The total ozone optical depth was determined only from the DS measurements. For that reason, the rising trend of total ozone optical depth for the monitored wavelengths in the period of 1994–2016 was equally statistically insignificant. For the wavelength of 306.3 nm, the value of the trend is 0.005 ±0.009 for 10 years, and the value of the trend for the wavelength of 320 nm is 0.001 ±0.002 for 10 years.

Figure 9 shows annual averages of AOD for the wavelength of 320 nm together with the uncertainty of their determination. The annual averages of AOD were calculated in a standard manner, i.e., by means of an average value of ETC. The lower limit of uncertainty was calculated by means of an average value of ETC, from which its standard deviation for the given intercalibration period was deducted. The upper limit of uncertainty was determined by analogy. The range of uncertainty interval depends primarily on suitable weather conditions in the given intercalibration period, as well as on the stability and homogeneity of measurements on days when it was possible to determine the ETC. The number of days, when it was possible to determine the ETC, plays its role too. For instance, there were only two measurements in the first intercalibration period (it covers 1994 and a smaller part of 1995), and for that reason, inter alia, the uncertainty interval for 1994 is narrow and has a very low relevance. In other intercalibration periods, there were at least 8 ETCs. Therefore, the following reliability intervals can be deemed trustworthy.

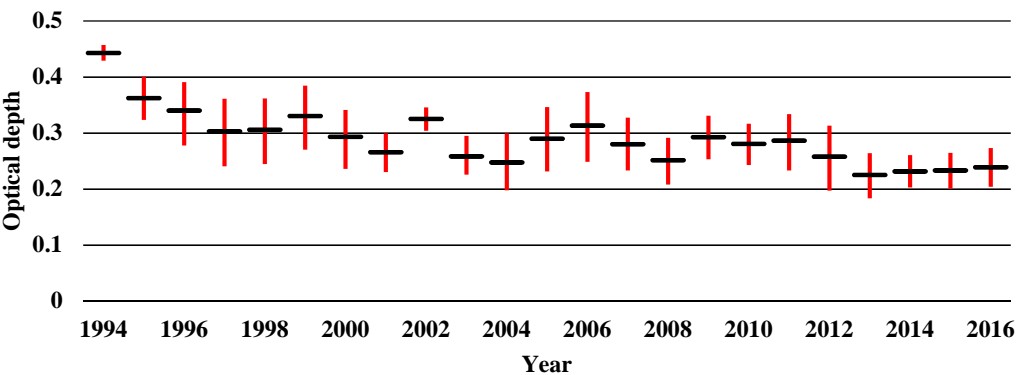

**Figure 9: Annual averages of AOD and their uncertainty for the wavelength of 320 nm, 1994–2016.**

Figure 10 shows a comparison of annual averages for the total ozone and aerosol optical depth for the selected wavelengths. It is obvious already at a glance that the AOD, unlike the total ozone optical depth, exhibits an apparent decline for the monitored period. For the wavelength of 306.3 nm, the value of the trend is –0.07 ±0.01 for 10 years, and the value of the trend for the wavelength of 320 nm is –0.06 ±0.01 for 10 years. If we omit first 2 years due to their lower reliability, then the value of the trend for the wavelength of 306.3 nm is –0.05 ±0.01 for 10 years, and the value of the trend for the wavelength of 320 nm is –0.04 ±0.01 for 10 years. In view of the above, a conclusion can be made that the transmittance of the atmosphere in the region of examined wavelengths has increased in the area of Poprad-Gánovce over the last 23 or 21 years. The total optical depth of the atmosphere for the wavelength of 306.3 nm has a trend with the value of –0.07 ±0.01 for 10 years, and its trend for the wavelength of 320 nm has the value of –0.06 ±0.01 for 10 years. If the first 2 years are omitted, the value of the trend for the wavelength of 306.3 nm is –0.06 ±0.01 for 10 years, and the value of the trend for the wavelength of 320 nm is –0.04 ±0.01 for 10 years. The negative trend has been caused particularly by the drop in the AOD.

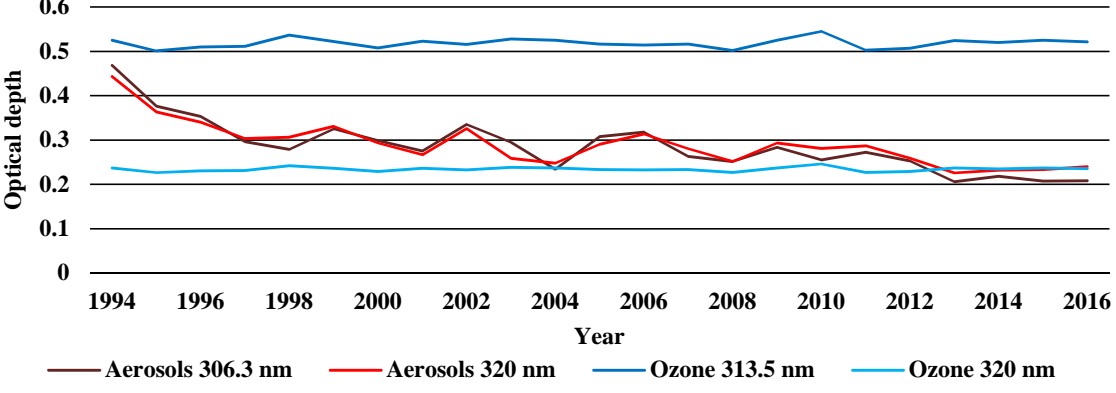

**Figure 10: Comparison of values of average annual optical depth for the selected wavelengths for ozone and aerosols, 1994–2016.**

Figure 11 presents a comparison of multiannual averages for the ozone optical depth, optical depth of Rayleigh scattering and AOD for all five examined wavelengths. It can be seen that ozone is dominant only in case of the shortest wavelength. It is exactly the reverse for the pair of the longest wavelengths, and ozone has the lowest impact among the examined factors. Rayleigh scattering has a dominant position in attenuation of direct solar radiation for all wavelengths except for the shortest one. In case of AOD, no unambiguous dependence of its size on the wavelength was observed. With respect to the long-term average, the AOD reached the highest value for the wavelengths of 310 nm and 316.8 nm, namely 0.32. On the contrary, the AOD reached the lowest value for the wavelengths of 306.3 nm and 320 nm, namely 0.29. The middle wavelength of 313.5 nm is characterized by the value of 0.3. With regard to the long-term average for 23 years, the difference in AOD for the shortest and the longest wavelengths is only –0.004. For the test year of 2014, this difference was –0.014. In conclusion, it can be stated that values of ETC have probably the greatest impact on the observed difference, while the value of the observed difference in AOD is consequently derived from them. It is also substantiated by Fig. 10. It can be seen that the AOD is greater for the wavelength of 306.3 nm than for the wavelength of 320 nm in certain years, while it is the reverse in other years.

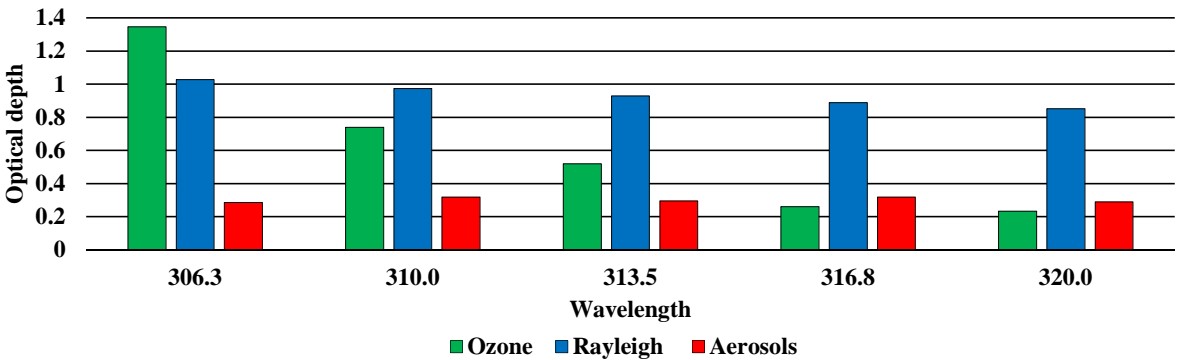

**Figure 11: Comparison of long-term averages for the total ozone optical depth, the optical depth of Rayleigh scattering and AOD, 1994–2016.**

Figure 12 illustrates an annual cycle of total ozone optical depth and its variability for individual months for the wavelength of 320 nm. The optical depth reaches its maximum in April with the value of 0.263 and the minimum in October with the value of 0.202. The highest variability is attributed to the month of February and the lowest one to the month of July. It applies to all the months of the year that the variability is considerably lower compared to the average. The month of February has the maximum variation coefficient amounting to 5.9 %. All the aforementioned characteristics of the total ozone optical depth depend only on the TCO value. It is peculiar to the central European location of the station that the annual maximum of the TCO occurs in April and the annual minimum occurs in October. It is also confirmed by the measurements from the nearby station in Hradec Králové (Vaníček et al., 2012). Therefore, it can be stated that the observed annual course of the total ozone optical depth is typical for the central European location of the station.

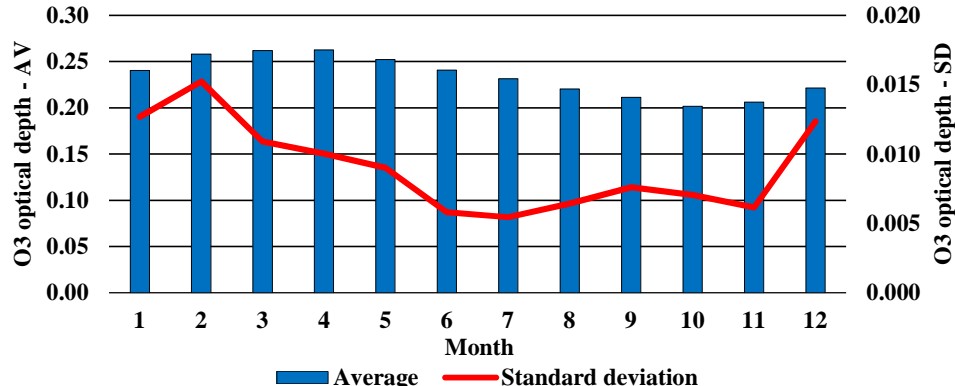

**Figure 12: Average monthly characteristics of the total ozone optical depth for the wavelength of 320 nm, 1994–2016.**

Figure 13 presents the same characteristics as in Fig. 12, but they refer to the AOD in this case. The annual cycle of monthly
5   averages is characterized by two peaks. The primary peak, that is the annual maximum, occurs in the month of August with
the value of 0.41. The secondary peak occurs in the month of April with the value of 0.4. The minimum is in the month of
December with the value of 0.16. The variability is also characterized by a two-peak annual cycle. The primary and
secondary peaks are in the same months as in case of the average. The annual minimum occurs in the month of November.
The variation coefficient reached significantly higher values compared to ozone. The minimum is attributed to the month of
10   May amounting to 19.1 %. The maximum is attributed to the month of December amounting to 42.2 %. The location of the
station at a higher altitude in the submontane area has the primary impact on the said characteristics, which explains lower
values of AOD in winter months. Higher values of AOD in spring and summer months are presumably related to agricultural
activities in the vicinity of the station.

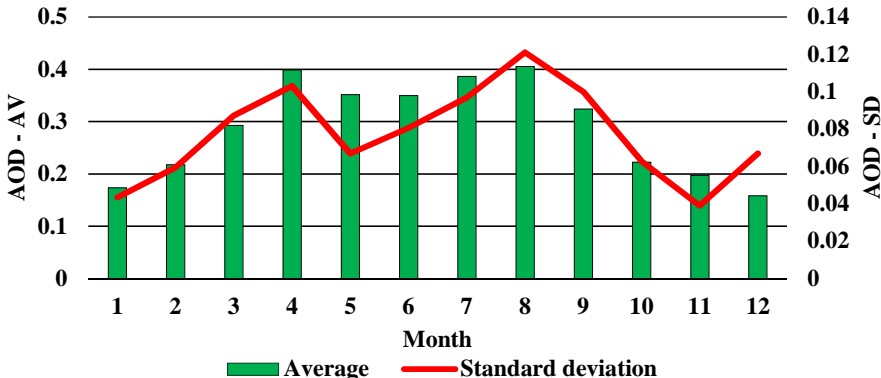

15   **Figure 13: Average monthly characteristics of the AOD for the wavelength of 320 nm, 1994–2016.**

Figure 14 shows the relative distribution of daily AOD averages for individual months for the monitored period of
1994–2016. It is important to point out that only the days, for which it was possible to determine the AOD, were taken into

consideration. In the month of December, there is the highest percentage of days with AOD within the interval from 0 to 0.1, specifically up to 40 %. On the contrary, the lowest percentage was seen in the month of August with the value of only 1 %. Days with a daily average of AOD above 0.3 dominate in the months of April to August. Days with a daily average of AOD within an interval below 0.3 dominate in the remaining months of the year. The AOD above 0.6 occurs the most often in the

month of August, that is in 18 % of days. The lowest percentage of such days is in the month of November and December, namely only 1 %. In conclusion, it can be stated that the presented percentages of individual days correlate very well with the annual cycle of AOD in Fig. 13.

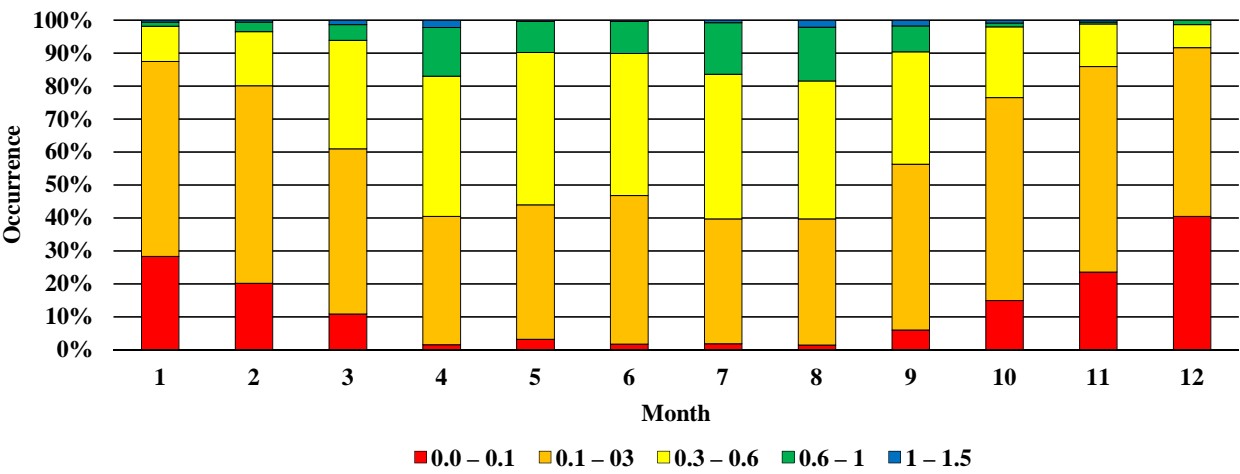

**Figure 14: Relative distribution of daily average values of AOD for the wavelength of 320 nm, 1994–2016.**

**4 Conclusions**

The total ozone and aerosol optical depth were determined by the Brewer ozone spectrophotometer in Gánovce near Poprad. The AOD was calculated by means of ETCs obtained using the Langley plot method. The analysed data of the direct solar radiation were available from 1994 to 2016, which is a 23-year series of measurements. In the 1990s, the ETCs were not determined during calibrations. Therefore, the LPM is the only practicable method for the determination of AOD for the

whole series of measurements. The ETCs used in this paper were determined for the 2-year intercalibration period. Use of such a long time period has both its advantage and disadvantage. The advantage lies in the fact that the ETC determined as an average of a higher number of days prevents the accidental impact of fluctuations that have no particular explanation. The disadvantage lies in the suppression of weather influences (especially the change of temperate) and modifications of the instrument in time intervals of less than 2 years. In addition to the methodology of ETC determination, the employed cloud

screening affects the values of AOD as well. If the cloud screening was not employed, the values of AOD would have been distinctly higher. Cloud screening employed in this paper is relatively simple. Hence, it can be argued that the actual AOD is even somewhat lower. It pertains particularly to the presented monthly and annual averages. In the future, the cloud

screening methodology is planned to be improved. As a result, it will be possible to obtain more representative values of AOD.

On the basis of a comparison of the Brewer spectrophotometer measurements using the Langley plot method and the Brewer software method, which is a part of the Operational Brewer Program, with Cimel sunphotometer measurements, it was
determined that the LPM achieved better results of the two compared methods. It was also determined that the application of corrections to the diffuse radiation, stray-light effect and polarization decreased the difference between the value of AOD for the shortest and longest of examined wavelengths. Although the difference has been reduced, it still reaches negative values, which results in the negative values of the Ångström exponent as well. With regard to the long-term average for 23 years, the difference in AOD is only –0.004. The application of correction for the polarization had the most significant influence on the
reduction of the given difference. On the contrary, the lowest contribution was achieved by the correction with regard to the diffuse radiation. The key factor influencing the value of the examined difference is probably the size and accuracy of ETC determination.

The obtained results clearly show the decrease in average annual values of the total optical depth of the atmosphere for the monitored wavelengths from 1994 to 2016. The slightly upward trend of the total ozone has been observed, which is
exhibited by a statistically insignificant increase in the total ozone optical depth. This insignificant increase has only a minimum effect on the trend of the total optical depth of the atmosphere. The root cause of the decrease in the total optical depth of the atmosphere is the statistically significant drop in the AOD. It follows that the transmittance of the atmosphere in the UV region of the spectrum from 306.3 nm to 320.1 nm has increased in the location of Poprad-Gánovce.

*Data availability*

The data used in the research and presented in this paper are available from the author upon agreement. It is recommended to send a request by email.

*Competing interests.* The author declares that he has no competing interests.

*Acknowledgements.* The author of the paper thanks the Slovak Hydrometeorological Institute for the provision of data. A special acknowledgement goes to Anna Pribullová for running the station of the Aerological and Radiation Centre, SHMI, in Poprad-Gánovce. The author also thanks the Faculty of Mathematics, Physics and Informatics of Comenius University in Bratislava for the possibility of external PhD study in meteorology and climatology, during which this paper was prepared.
Many thanks also to the organizing team of international Quadrennial Ozone Symposium held in Edinburgh on 4–9 September 2016, namely, inter alia, for the provision of financial support. Finally, the author thanks the European Aerosols, Clouds, and Trace gases Research Infrastructure (ACTRIS) for providing the calibration of the Cimel sunphotometer.

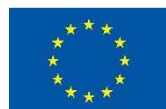 ACTRIS-2 project has received funding from the European Union's Horizon 2020 research and innovation programme under grant agreement No 654109.

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
