# Peer review of "Comparison of the optical depth of total ozone and atmospheric aerosols in Poprad-Gánovce, Slovakia"

_Atmospheric Chemistry and Physics, 2017_

## Referee Comment (RC1) · Anonymous Referee #2 · 2 Jun 2017

1) General comments

The present paper includes a long UV AOD series spanning from 1994 to 2016 which may be useful to provide further insight on the role of aerosols on the Earth's climate. Furthermore, these data may be also used to demonstrate the capability of the Brewer spectrophotometer to measure AOD, most of these instruments being used only for ozone measurements.

In my opinion, these two points make the paper interesting for the scientific community. There are however three main issues:

a) There is still room to improve the scientific discussion, see points 2a-i below

[Figure]

b) It is extremely important to show that the Brewer AOD is correct. For that, you need to provide meaningful comparisons with data from other instruments, see points 2j-k.

c) The quality of the presentation and, specially, of the English has to be improved, see Section 3 below.

Without improvements in these three areas, I can not support the publication of the paper in ACP. With a view to help the author improve the paper, I will provide specific questions and comments next.

2) Scientific discussion

a) On page 3, line 16, the author states that "In this case, the AOD calculation algorithm is part of the main control program for Brewer. The main difference from previous method is that the ETCs for individual wavelengths are not determined by LPM method but they are obtained during calibrating the instrument, i.e. every 2 years". Could the author explain how the ETCs are determined during this calibration of the instrument? Have you compared the results from the LPM and the so-called calibration methods?

b) On page 7, Fig. 1, there is an entry with the label "AOD-AAOD<0.5". This part of the screening algorithm does not seem to be explained in the text, and the definition of "AAOD" seems to be missing from the paper.

c) On page 7, line 13, the author states that "Daily averages are calculated as arithmetic average of all values of a given day (from at least one value)." So, a daily average is considered valid even if there is just one AOD value for the day? How many AOD measurements do you obtain on average for each day? How many times do you get just one measurement in a day?

d) From the discussion on page 8 about the calibration periods, it is not clear how they are selected. Are they the same as the period between the standard ozone calibrations? If so, does the author find that the stability of the ozone and AOD configurations are the same? Did the author try to use shorter calibration periods?

e) On Page 9, line 14, the author mentions that "ground measurements from the nearby station in Hradec Králové and satellite data" were used to complete the series down to 1962. What type of ground instrument operates at Hradec Králové? What satellite data was used? Could the author show the full series from 1962 to 2016?

f) From Figs. 2 and 4 and text, it's clear that the year 1994 is very noisy for the AOD data. Was this year included in the analyses (specially, the determination of the trends) in Section 3.2?

g) Could the author provide some explanation for the behavior shown in Figs. 6 and 7? (E.g., are the peaks be related to weather conditions?)

h) What conclusions does the author extract from Fig. 8?

i) On page 13, line 20, the author states that "Such a significant difference is caused by inconsistent methodology for the calculation of total ozone through ZS measurements". Could the author elaborate further why does he consider the ZS measurement method inconsistent?

j) On page 13, line 27, while comparing the Brewer and OMI AOD data, the author writes that "The correlation coefficient has reached the value of 0.51 in comparison with each other what represents a strong positive correlation". I don't believe 0.51 should be considered a strong correlation. For comparison, what's the correlation coefficient between the Brewer and OMI ozone data? Instead of the bar plots in Fig. 9, could you plot the fits as in Fig. 10?

k) As stated above, showing the readers that your Brewer AOD is correct is of the utmost importance. The comparison with the satellite data, as shown in this work and others before, sometimes might not be straightforward. Making a comparison with other ground-based instruments would be thus a better option. According to the AERONET database, there is level 2.0 Cimel AOD data at 340 nm for the Poprad-Gánovce site from December 2014 to January 2017, see

https://aeronet.gsfc.nasa.gov/cgi-bin/type_one_station_opera_v2_new?site=Poprad-Ganovce&nachal=0&year=22&aero_water=0&level=3&if_day=0&if_err=0&place_code=10&year_or_month=1

Could the author use these data to compare with the Brewer AOD? If not, could they provide a comparison with another ground-based instrument? If not, has the author considered the possibility of attending to some inter-comparison campaign?

3) Presentation

a) First and foremost, the quality of the English has to be improved. This is not a purely cosmetic question, there are sentences that are very difficult to understand, like e.g. "The lower limit of uncertainty was calculated using average value ETC from which its standard deviation has been deducted in the given calibration period" on page 9. Please, do check the whole paper and improve the English to an acceptable level.

b) As mentioned before, it is critical to demonstrate that the Brewer AOD is correct. Such proof should come immediately after the AOD calculation method is presented and before any other discussion of the data. I thus suggest inserting Sec 3.3 (now including a comparison with AERONET data) before Sec 3.1

---

## Referee Comment (RC2) · Anonymous Referee #1 · 5 Jun 2017

GENERAL COMMENTS

The manuscript by dr. Peter Hrabčák presents a long dataset of total ozone and aerosol optical depth (AOD) measurements taken by a Brewer spectrophotometer at the site of Poprad-Gánovce. The optical depth of these two atmospheric constituents are calculated and compared. A basic statistical analysis of the annual averages is presented, which supports a statistically significant decrease of the AOD. Comparisons with satellite estimates and zenith sky measurements are also briefly reported.

An in-depth analysis of such a long dataset (23 years) is certainly of interest and useful for the atmosphere and climate communities. However, I don't support the publication

of the manuscript until major revisions are made, in particular:

1. the language must be definitely improved for ease of reading;

2. the statistical analysis is too simplistic:

- an in-depth examination of the measurement uncertainty is essential to trust the quality of Brewer measurements and to correctly calculate the trend significance, however it is missing in the manuscript. The calculation of the statistical significance was performed on the basis of the natural/instrumental variability from the data themselves. Instead, systematic errors and drift uncertainties should be included in the calculation of the trend uncertainty and its statistical significance. Furthermore, the paper would certainly benefit from comparison to co-located (or close) photometers, if available;

- the observed decrease in the AOD series seems to be mainly introduced by the first two years of data (Fig. 3-4). Since "instability of ETCs during the 1st calibration period" is reported in Sect. 3.1 and the trend abruptly changes after 1996, I'm wondering if use of a single trend for the full series is justified and what the statistical significance of the trend would be by considering only the 1996-2016 period;

- the results of the analysis are not properly supported by a complete understanding and explanation of the physical phenomena at their base. The reported trends are not compared to the existing scientific bibliography.

These remarks are elaborated in the next sections (specific comments and technical corrections).

SPECIFIC COMMENTS

- Sect. 1 (p.3, l.5): the Langley plot method should be explained in Sect. 2.3 instead of the Introduction. Instead, a list of previous publications about AOD and Brewers and trend calculations would be welcome in Sect. 1;

- Sect. 2.1 should be rewritten in a more rigorous way. Use bibliographic references

and/or formulae instead of a qualitative description. Moreover, zenith sky measurements should be introduced and explained here and the ZS retrieval algorithm should be presented (how was the ZS polynomial determined?);

- Sect. 2.1: possible uncertainties due to the use of a single monochromator should be discussed. The sources of the deviations from the Angostrom law of the AOD measured by single Brewers at the shortest UV wavelengths (e.g., Fig. 5) are known since a long time (cf. Arola and Koskela (2004), "On the sources of bias in aerosol optical depth retrieval in the UV range"). Uncertainties arising from polarisation effects and temperature changes inside the instrument should also be mentioned;

- Sect. 2.3 and 2.4: some inconsistencies can be found in the use of different cross sections datasets in the manuscript. Effective IUP cross sections are used for the calculation of ozone optical depths, while ozone retrievals are performed using Bass&Paur (cf. Redondas et al., 2014). Bodhaine et al., 1999, is used for AOD calculations, while the Brewer operational Rayleigh cross sections are used to retrieve the ozone optical depth, causing a bias of about 3 DU (cf. Carlund et al., 2017). The authors should explore those issues;

- Sect. 2.4 (p.6, l.6-19): explain why all those thresholds were chosen. Explain that an iterative approach was used;

- Sect. 2.4 (p.7, l.5): the figure content should be fully explained in the text;

- Sect. 2.4 (p.7, l.13-19): with no thresholds for the minimum number of data for a month and a year, the annual average becomes very sensitive to potential gaps in the series and their distribution throughout the year (due to the seasonal cycles of both ozone and AOD). The author should explain how he dealt with gaps in the series. The calculation of the linear trend uncertainty as described in the manuscript only takes into account the natural and instrumental variability. However, also instrumental systematic errors (e.g., radiometric calibration drifts) contribute to the uncertainty of the measurements and the trend and should be taken into account to determine the trend

significance. An extensive treatment and description of measurement uncertainties is lacking in the current manuscript;

- Sect. 3.1: according to Fig. 2, the logarithm of ETC jumps by more than 0.2 at several points. The author should prove that a two-year "piecewise" series of calibration constant is suitable for the calculation and explain why a moving average wouldn't be better. "Instrument instability", "straylight effects" and "instrumental problems" reported in Sect. 3.1 should be better explained and quantified. Furthermore, could you plot the ETCs obtained by transfer from IOS in the same figure and check whether they agree with the values obtained from LPM?

Also, Langley plots in urban areas are prone to errors: the ETC variability could originate from AOD curvature centred at noon, which cannot be filtered by any quality criteria (e.g., Marenco 2007 and Diémoz et al. 2016). Are in-situ measurements (e.g., PM) available in the investigation area to exclude such an effect?

- Sect. 3.2: the AOD trend in the first three years of measurement is about -0.1/year (Fig. 3). Does the author have a reasonable explanation for this large decrease? A reference to Arola and Koskela (2004) should be included to explain the observed AOD dependence on wavelength. The seasonal variability of ozone is a well-known phenomenon, however the general behaviour drawn in Fig. 6 should be better explained in the text and appropriate references provided. Similarly, the AOD seasonal cycle (Fig. 7) should be explained, the physical reasons for the observed two peaks searched for and the general behaviour in Fig. 8 explained. The AOD results should be compared to analogous data already published in the scientific literature;

- Sect. 3.3: instantaneous measurements from the Brewer should be compared to overpassing satellite estimates instead of daily means. Also, why DS and ZS are not directly compared? This would answer the question raised in the Conclusions ("The reason for this is _probably_ the systematic error of the ozone determination using ZS measurements"). The data selection criteria for the OMI-Brewer comparison (223

days) should be better explained (distances in space and time to Brewer measurements). How were the satellite data obtained (e.g., GIOVANNI)?

- Conclusions: the match between OMI and the Brewer is defined "very good", however the linear correlation index is only 0.5. Some bibliographic references should be provided to prove that the agreement between both instruments is satisfactory compared to similar data in the existing scientific literature;

TECHNICAL CORRECTIONS

- p.1, l.17: replace "terrestrial" with "ground-based" throughout the manuscript. The name of the satellite radiometer (OMI) should be specified in the abstract. Some statistical scores should be included in the abstract to quantitatively support the "very good match" claim;

- p.1, l.19 "systematically higher values": the value of the bias should be written;

- p.1, l.26 "Adverse effects have higher doses of UV radiation...": the words order is wrong;

- p.2, l.1 "Very necessary" -> "Necessary";

- p.2, l.2 "functioning of the human body": be more specific. "The anthropogenic effect": what effect? Please, reformulate the sentence;

- p.2, l.5 "about 5%": specify the considered latitude belt. "even lower": why "even"?

- p.2, l.17: "it affects" -> "they affect". "ones" -> "one". "aircraft flight" -> "emission from aircrafts";

- p.2, l.20: "the significant" -> "a significant";

- p.2, l.22-32: I would remove this paragraph, which is too didactic and a bit off topic, since bibliographic references were already introduced;

- p.2, l.33: "have a significant role to play" -> "play a significant role";

Interactive
comment

[Figure]

- p.3, l.1: it should be better explained that AOD is not the only quantity describing the radiative effects of aerosols;

- p.3, l.2: "AOD obtained results... and satellite measurements" please rephrase the sentence;

- p.3, l.4: "Brewer allows" -> "Brewer spectrophotometers allow". "Optical depth" of what?

- p.3, l.8 "there is required...": rephrase the whole sentence;

- p.3, l.10: "zenith angles" -> "solar zenith angles";

- p.3, l.12: it should be explained why lower latitudes are better and what are the "certain limitations in middle and ... higher latitudes";

- p.3, l.17: "previous method" -> "the previous method";

- p.3, l.23: "The Brewer ozone spectrophotometer";

- p.3, l.25: "different" from what?. "after pass" -> "after passing";

- p.3, l.26: notice that the DOAS method refer to continuous spectral measurements nowadays, while the Brewer only measure irradiance at five wavelengths;

- p.3, l.28: "predetermined wavelengths" -> what wavelengths exactly, and how many?

- p.3, l.29: "It is possible to determine the total amount...";

- p.3, l.30 "by following a comparative analysis in the mathematical model...": the sentence is totally obscure to the reader that doesn't know how a Brewer works;

- p.3, l.32 "feasible": explain why it is possible only at those wavelengths;

- p.4, l.1 "0.006+- 0.002 nm": notice that this is the wavelength increment (1 microstep) rather than the accuracy, which depends on temperature changes inside the instrument and frequency of hg tests;

- p.4, l.2: "undergoing" -> "has been undergoing";

- p.4 l.6: Sect. 2.2 should be the first section, in order to keep the description of the instrument (now 2.1) and of the algorithm (2.3) close to each other;

- p.4, l.9 "The content of aerosol in the air, whether...": please rephrase this sentence, which is not clear;

- p.4, l.11 "In rare cases, it can also be...": rephrase;

- p.4, l.14 "relatively windy": relatively in comparison to what?. A figure with a map with the position of the site (e.g., including a wind rose) would be helpful;

- p.4, l.17 "Measured total ozone values... we used...": wrong words order;

- p.4, l.19 "density" -> "power density" or "irradiance";

- p.5, l.1 "It is used to using so called an effective": rephrase;

- p.5, l.10 "Rayleight scattering" -> "Rayleigh scattering". Please, specify here which dataset was used, e.g. Bodhaine 1999;

- p.5, l.17-22: the description of the Brewer data reduction is not clear at all to the unaccustomed reader. Please, rewrite this part or replace with bibliographic references;

- p.5, l.23 "the arithmetic average is then calculated": how is the effective airmass for the average of those 5 measurements calculated? Remember that air mass doen't vary linearly with time;

- p.5, l.25: "zeniths" -> "zenith angles";

- p.5, l.29: "angels" -> "angles";

- p.5, l.30: "determine unknown" -> "determine the unknowns";

- p.6, l.1-5: rewrite this paragraph in a more ordered way;

- p.6, l.30: provide a reference for the airmass formula;

- p.7, l.7: "received" -> "ended with";

- p.7, l.13 "characteristics ... progressed": the sentence is unclear;

- p.7, l.18: "standard deviation" of which quantity?

- p.8, l.8 "is the highest for the shortest wavelength": rephrase. "Variation coefficient" -> "The variation coefficient";

- p.8, l.11: "A graph also shows ETCs values which characterize entire" -> "The graph also shows the ETCs values which characterize the entire";

- p.8, l.12 "25 ETCs were used on average": 25 in a year?. "Strict conditions met" -> "were met";

- p.8, l.15: "the chosen criteria are the optimal compromise". "Directly affects the resulting values";

- p.8, l.16: what are the "weather effects"? Does the author refer to the clouds?

- p.9, l.14-18: provide bibliographic references of previous studies;

- p.9, l.17 and 18: "what" -> "which". Line 21: "than" -> "that". Lines 21-23: several "the" are missing;

- p.9, l.24: "the comparison". L.27: "the mentioned". L.28: "in studied" -> "at the studied";

- p.10 Fig.3: use the same wavelengths for ozone and AOD optical depths;

- p.11-12, Fig. 6-7: draw AOD and its standard deviation on the same scale, e.g. using boxplots;

- p.11, l.19: "it occurs" -> "occurs". Line 20: "characteristic" -> "characterized";

- p.12, l.12: "in particular";

- p.13, l.8 "once a day": provide overpass time;

- p.13, l.9 "scored specific place": rephrase. Line 10: "the square" -> "a square";

- p.13, l.13: "is illustrated comparison of the annual averages..." -> "the comparison of the annual averages of total ozone is illustrated in Fig. 9". "it was the comparison of values obtained ... obtained": rephrase;

- p.13, l.20: is 3.9 DU a "significant" difference? Line 21: "It can be said that...": please, explain better which sources of systematic errors have to be considered;

- p.13, l.24: what value for the "Angstrom exponent" was used? Why?

- p.13, l.27 "only AOD values smaller than 1.5": how was this threshold established?

- p.13, l.28: I would say that a correlation index of 0.5 is moderate, not strong;

- p.14, l.10: "was" -> "were". "so it is a period" -> "i.e. for a period".

---

## Author Comment (AC1) · 4 Jul 2017

Thank you very much for your comments.

I have been working on a new version of my manuscript since june. I am trying to take into account must of your comments. The revised manuscript will be presented probably in August.

---

## Author Response (AR1)

**Author's response to the comments of Anonymous Referee #1 and #2.**

**First of all, I would like to apologize for my delay.**

Author's response to the comments of Anonymous Referee #1. - bellow

Author's response to the comments of Anonymous Referee #2. - from page 12

**Author's response to the comments of Anonymous Referee #1.**

**Authors comments are written in bold.**

GENERAL COMMENTS

The manuscript by dr. Peter Hrabčák presents a long dataset of total ozone and aerosol optical depth (AOD) measurements taken by a Brewer spectrophotometer at the site of Poprad-Gánovce. The optical depth of these two atmospheric constituents are calculated and compared. A basic statistical analysis of the annual averages is presented, which supports a statistically significant decrease of the AOD. Comparisons with satellite estimates and zenith sky measurements are also briefly reported.

An in-depth analysis of such a long dataset (23 years) is certainly of interest and useful for the atmosphere and climate communities. However, I don't support the publication of the manuscript until major revisions are made, in particular:

1. the language must be definitely improved for ease of reading;

2. the statistical analysis is too simplistic:

- an in-depth examination of the measurement uncertainty is essential to trust the quality of Brewer measurements and to correctly calculate the trend significance, however it is missing in the manuscript. The calculation of the statistical significance was performed on the basis of the natural/instrumental variability from the data themselves. Instead, systematic errors and drift uncertainties should be included in the calculation of the trend uncertainty and its statistical significance. Furthermore, the paper would certainly benefit from comparison to co-located (or close) photometers, if available;

**The manuscript was significantly redesigned. Section 3.1 Comparison of AOD values obtained by LPM, BSM and Cimel sunphotometer and section 3.3 Correction for the diffuse radiation, stray-light effect and polarization were added, among other things.**

- the observed decrease in the AOD series seems to be mainly introduced by the first two years of data (Fig. 3-4). Since "instability of ETCs during the 1st calibration period" is reported in Sect. 3.1 and the trend abruptly changes

after 1996, I'm wondering if use of a single trend for the full series is justified and what the statistical significance of the trend would be by considering only the 1996-2016 period;

**Among other things, this answer is included in the manuscript (page 19 , line 15): It is obvious already at a glance that the AOD, unlike the total ozone optical depth, exhibits an apparent decline for the monitored period. For the wavelength of 306.3 nm, the value of the trend is –0.07 ±0.01 for 10 years, and the value of the trend for the wavelength of 320 nm is –0.06 ±0.01 for 10 years. If we omit first 2 years due to their lower reliability, then the value of the trend for the wavelength of 306.3 nm is –0.05 ±0.01 for 10 years, and the value of the trend for the wavelength of 320 nm is –0.04 ±0.01 for 10 years.**

- the results of the analysis are not properly supported by a complete understanding and explanation of the physical phenomena at their base.

**The part Methodology and the part Results and discussion was redesigned.**

The reported trends are not compared to the existing scientific bibliography.

**Manuscript contains this part (page 2, line 11): Anthropogenic emissions of aerosols have been gradually reduced in the developed countries, and a drop in the aerosol optical depth (AOD) has been observed in several locations (Kazadzis et al., 2007; Mishchenko and Geogdzhayev, 2007; Alpert et al., 2012; de Meij et al., 2012; Zerefos et al., 2012). Moreover, introduction was enriched by the examples of trend calculations in the other places.**

These remarks are elaborated in the next sections (specific comments and technical corrections).

SPECIFIC COMMENTS

- Sect. 1 (p.3, l.5): the Langley plot method should be explained in Sect. 2.3 instead of the Introduction.

**The explanation of the Langley plot method was relocated to Sect. 2.4 instead of the Introduction.**

Instead, a list of previous publications about AOD and Brewers and trend calculations would be welcome in Sect. 1;

**It was accepted. Please, see section 1.**

- Sect. 2.1 should be rewritten in a more rigorous way. Use bibliographic references and/or formulae instead of a qualitative description.

**The section has been rewritten, but now the content is in the section 2.2**

Moreover, zenith sky measurements should be introduced and explained here and the ZS retrieval algorithm should be presented (how was the ZS polynomial determined?);

**The issue of ZS measurements has been removed from the manuscript. It was done because this issue is complicated and the scope of manuscript is moreover quite big.**

- Sect. 2.1: possible uncertainties due to the use of a single monochromator should be discussed. The sources of the deviations from the Angostrom law of the AOD measured by single Brewers at the shortest UV wavelengths (e.g., Fig. 5) are known since a long time (cf. Arola and Koskela (2004), "On the sources of bias in aerosol optical depth retrieval in the UV range").

**It was accepted. Please, see section 2.2.**

Uncertainties arising from polarisation effects and temperature changes inside the instrument should also be mentioned;

**It was accepted. Please, see section 2.2.**

- Sect. 2.3 and 2.4: some inconsistencies can be found in the use of different cross sections datasets in the manuscript. Effective IUP cross sections are used for the calculation of ozone optical depths, while ozone retrievals are performed using Bass&Paur (cf. Redondas et al., 2014). Bodhaine et al., 1999, is used for AOD calculations, while the Brewer operational Rayleigh cross sections are used to retrieve the ozone optical depth, causing a bias of about 3 DU (cf. Carlund et al., 2017). The authors should explore those issues;

**The total column of ozone was recalculated by Bodhaine et al., 1999 and IUP. Now, it is the same like for the optical depth calculations. This issues were explored in section 2.3 and section 2.4.**

- Sect. 2.4 (p.6, l.6-19): explain why all those thresholds were chosen. Explain that an iterative approach was used;

**Section 2.4 was sophisticated, but it is not possible to explain all things in depth because the manuscript is quite big.**

- Sect. 2.4 (p.7, l.5): the figure content should be fully explained in the text;

**Explanation of cloud screening is now included in the manuscript (page 10).**

- Sect. 2.4 (p.7, l.13-19): with no thresholds for the minimum number of data for a month and a year, the annual average becomes very sensitive to potential gaps in the series and their distribution throughout the year (due to the seasonal cycles of both ozone and AOD). The author should explain how he dealt with gaps in the series.

**During operation of Brewer spectrophotometer did not appear big gap in data. Instrument is in standard operating every day, if possible.**

The calculation of the linear trend uncertainty as described in the manuscript only takes into account the natural and instrumental variability. However, also instrumental systematic errors (e.g., radiometric calibration drifts) contribute to the uncertainty of the measurements and the trend and should be taken into account to determine the trend significance. An extensive treatment and description of measurement uncertainties is lacking in the current manuscript;

**The section 3.3 Correction for the diffuse radiation, stray-light effect and polarization was added.**

- Sect. 3.1: according to Fig. 2, the logarithm of ETC jumps by more than 0.2 at several points. The author should prove that a two-year "piecewise" series of calibration constant is suitable for the calculation and explain why a moving average wouldn't be better.

**The moving average probably is not a good idea because a variability of ETCs in the intercalibration period is too high due to inaccuracy of ETC determination and the number of determined ETCs is too low. The two-year interval was set also due to a comparisom with Brewer softwere method. This answer is also included in the manuscript (page 15 , line 3): To calculate the ETC characterizing the entire 2-year period, 17 values of individual ETCs were employed with respect to the long-term average. This number is the same in case of all wavelengths. If the conditions were less strict, there would have been more days, for which it was possible to determine the ETC. On the other hand, the spread of determined ETCs would be wider, which would have a negative effect on the required accuracy. Therefore, the chosen criteria represent an optimum compromise.**

"Instrument instability", "straylight effects" and "instrumental problems" reported in Sect. 3.1 should be better explained and quantified.

**It was partially accepted. Please, see section 3.3.**

Furthermore, could you plot the ETCs obtained by transfer from IOS in the same figure and check whether they agree with the values obtained from LPM?

**Values of ETCs were compared. Please, see section 3.2.**

Also, Langley plots in urban areas are prone to errors: the ETC variability could originate from AOD curvature centred at noon, which cannot be filtered by any quality criteria (e.g., Marenco 2007 and Diémoz et al. 2016). Are in-situ measurements (e.g., PM) available in the investigation area to exclude such an effect?

**It is not possible to investigate PM measurements.**

- Sect. 3.2: the AOD trend in the first three years of measurement is about -0.1/year (Fig. 3). Does the author have a reasonable explanation for this large decrease?

**It is probably link with the decrease in ETC values. Meaby there are also another influences, for example: natural variability or a decrease in combustion of coal and wood in the area of interest.**

A reference to Arola and Koskela (2004) should be included to explain the observed AOD dependence on wavelength.

**It was accepted. Please, see section 2.2.**

The seasonal variability of ozone is a well-known phenomenon, however the general behaviour drawn in Fig. 6 should be better explained in the text and appropriate references provided. Similarly, the AOD seasonal cycle (Fig. 7) should be explained, the physical reasons for the observed two peaks searched for and the general behaviour in Fig. 8 explained.

**It was partially accepted. Please, see section 3.4.**

The AOD results should be compared to analogous data already published in the scientific literature;

**Manuscript contains this part (page 12, line 32):** Pribullová (2002) does not mention the unambiguous dependence of AOD on the wavelength as well. It indicates the lowest AOD in case of the lowest wavelength and presents the highest values of AOD for the wavelength of 310 nm.

- Sect. 3.3: instantaneous measurements from the Brewer should be compared to overpassing satellite estimates instead of daily means. Also, why DS and ZS are not directly compared? This would answer the question raised in the Conclusions ("The reason for this is _probably_ the systematic error of the ozone determination using ZS measurements"). The data selection criteria for the OMI-Brewer comparison (223 days) should be better explained (distances in space and time to Brewer measurements). How were the satellite data obtained (e.g., GIOVANNI)?

**The issue of ZS measurements and issue of satellite measurements have been removed from the manuscript. It was done because this issue is complicated and the scope of manuscript is moreover quite big.**

- Conclusions: the match between OMI and the Brewer is defined "very good", however the linear correlation index is only 0.5. Some bibliographic references should be provided to prove that the agreement between both instruments is satisfactory compared to similar data in the existing scientific literature;

**The issue of comparisons with satellite measurements has been removed from the manuscript.**

TECHNICAL CORRECTIONS

- p.1, l.17: replace "terrestrial" with "ground-based" throughout the manuscript. The name of the satellite radiometer (OMI) should be specified in the abstract. Some statistical scores should be included in the abstract to quantitatively support the "very good match" claim;

**It is not needed, because the satellite part was removed.**

- p.1, l.19 "systematically higher values": the value of the bias should be written;

**It is not needed, because the ZS was removed.**

- p.1, l.26 "Adverse effects have higher doses of UV radiation...": the words order is wrong;

**Manuscript was translated by professional interpreter.**

- p.2, l.1 "Very necessary" -> "Necessary";

**Manuscript was translated by professional interpreter.**

- p.2, l.2 "functioning of the human body": be more specific. "The anthropogenic effect": what effect? Please, reformulate the sentence;

**Manuscript was translated by professional interpreter.**

- p.2, l.5 "about 5%": specify the considered latitude belt. "even lower": why "even"?

**It is clear now.**

- p.2, l.17: "it affects" -> "they affect". "ones" -> "one". "aircraft flight" -> "emission from aircrafts";

**Manuscript was translated by professional interpreter.**

- p.2, l.20: "the significant" -> "a significant";

**Manuscript was translated by professional interpreter.**

- p.2, l.22-32: I would remove this paragraph, which is too didactic and a bit off topic, since bibliographic references were already introduced;

**It has been let, but the introduction was enriched by another important things.**

- p.2, l.33: "have a significant role to play" -> "play a significant role";

**Manuscript was translated by professional interpreter.**

- p.3, l.1: it should be better explained that AOD is not the only quantity describing the radiative effects of aerosols;

**It has not been accepted.**

- p.3, l.2: "AOD obtained results... and satellite measurements" please rephrase the sentence;

**Manuscript was translated by professional interpreter.**

- p.3, l.4: "Brewer allows" -> "Brewer spectrophotometers allow". "Optical depth" of what?

**Manuscript was translated by professional interpreter.**

- p.3, l.8 "there is required...": rephrase the whole sentence;

**Manuscript was translated by professional interpreter.**

- p.3, l.10: "zenith angles" -> "solar zenith angles";

**Manuscript was translated by professional interpreter.**

- p.3, l.12: it should be explained why lower latitudes are better and what are the "certain limitations in middle and ... higher latitudes";

**In this part are the scientific literature references. In the references can be find details.**

- p.3, l.17: "previous method" -> "the previous method";

**Manuscript was translated by professional interpreter.**

- p.3, l.23: "The Brewer ozone spectrophotometer";

**Manuscript was translated by professional interpreter.**

- p.3, l.25: "different" from what?. "after pass" -> "after passing";

**Manuscript was translated by professional interpreter.**

- p.3, l.26: notice that the DOAS method refer to continuous spectral measurements nowadays, while the Brewer only measure irradiance at five wavelengths;

**The sentence has been removed.**

- p.3, l.28: "predetermined wavelengths" -> what wavelengths exactly, and how many?

**Section 2.2 was rewritten.**

- p.3, l.29: "It is possible to determine the total amount...";

**Manuscript was translated by professional interpreter.**

- p.3, l.30 "by following a comparative analysis in the mathematical model...": the sentence is totally obscure to the reader that doesn't know how a Brewer works;

**The sentence has been removed.**

- p.3, l.32 "feasible": explain why it is possible only at those wavelengths;

**Now, it is clear from previous sentences.**

- p.4, l.1 "0.006+- 0.002 nm": notice that this is the wavelength increment (1 microstep) rather than the accuracy, which depends on temperature changes inside the instrument and frequency of hg tests;

**It was accepted. Please, see section 2.2.**

- p.4, l.2: "undergoing" -> "has been undergoing";

**Manuscript was translated by professional interpreter.**

- p.4 l.6: Sect. 2.2 should be the first section, in order to keep the description of the instrument (now 2.1) and of the algorithm (2.3) close to each other;

**It was accepted.**

- p.4, l.9 "The content of aerosol in the air, whether...": please rephrase this sentence, which is not clear;

**Manuscript was translated by professional interpreter.**

- p.4, l.11 "In rare cases, it can also be...": rephrase;

**Manuscript was translated by professional interpreter.**

- p.4, l.14 "relatively windy": relatively in comparison to what?. A figure with a map with the position of the site (e.g., including a wind rose) would be helpful;

**It has been rewritten.**

- p.4, l.17 "Measured total ozone values... we used...": wrong words order;

**Manuscript was translated by professional interpreter.**

- p.4, l.19 "density" -> "power density" or "irradiance";

**Manuscript was translated by professional interpreter.**

- p.5, l.1 "It is used to using so called an effective": rephrase;

**Manuscript was translated by professional interpreter.**

- p.5, l.10 "Rayleight scattering" -> "Rayleigh scattering". Please, specify here which dataset was used, e.g. Bodhaine 1999;

**It is explained in the manuscript (page 9, line 28).**

- p.5, l.17-22: the description of the Brewer data reduction is not clear at all to the unaccustomed reader. Please, rewrite this part or replace with bibliographic references; -

**It was partially rewrite.**

 p.5, l.23 "the arithmetic average is then calculated": how is the effective airmass for the average of those 5 measurements calculated? Remember that air mass doesn't vary linearly with time;

**It was corrected. Now, air mass is calculated separately for all measurements (for every of five).**

- p.5, l.25: "zeniths" -> "zenith angles";

**Manuscript was translated by professional interpreter.**

 - p.5, l.29: "angels" -> "angles"; -

**Manuscript was translated by professional interpreter.**

p.5, l.30: "determine unknown" -> "determine the unknowns";

**Manuscript was translated by professional interpreter.**

- p.6, l.1-5: rewrite this paragraph in a more ordered way;

**It was partially rewrite.**

- p.6, l.30: provide a reference for the airmass formula;

**It was accepted, please see section 2.4.**

- p.7, l.7: "received" -> "ended with";

**Manuscript was translated by professional interpreter.**

- p.7, l.13 "characteristics ... progressed": the sentence is unclear;

**Manuscript was translated by professional interpreter.**

- p.7, l.18: "standard deviation" of which quantity?

**It is a standard deviation of a given linear trend.**

- p.8, l.8 "is the highest for the shortest wavelength": rephrase. "Variation coefficient" -> "The variation coefficient";

**Manuscript was translated by professional interpreter.**

- p.8, l.11: "A graph also shows ETCs values which characterize entire" -> "The graph also shows the ETCs values which characterize the entire";

**Manuscript was translated by professional interpreter.**

- p.8, l.12 "25 ETCs were used on average": 25 in a year?.

**For entire intercalibration period, it means 2 years.**

"Strict conditions met" -> "were met";

**Manuscript was translated by professional interpreter.**

- p.8, l.15: "the chosen criteria are the optimal compromise". "Directly affects the resulting values";

**Manuscript was translated by professional interpreter.**

- p.8, l.16: what are the "weather effects"? Does the author refer to the clouds?

**Yes, it means mainly clouds.**

- p.9, l.14-18: provide bibliographic references of previous studies;

**There is no references, it is my own calculations.**

- p.9, l.17 and 18: "what" -> "which". Line 21: "than" -> "that". Lines 21-23: several "the" are missing;

**Manuscript was translated by professional interpreter.**

- p.9, l.24: "the comparison". L.27: "the mentioned". L.28: "in studied" -> "at the studied";

**Manuscript was translated by professional interpreter.**

- p.10 Fig.3: use the same wavelengths for ozone and AOD optical depths;

**I think, it is not a good idea.**

- p.11-12, Fig. 6-7: draw AOD and its standard deviation on the same scale, e.g. using boxplots;

**I think, it is not a good idea.**

- p.11, l.19: "it occurs" -> "occurs". Line 20: "characteristic" -> "characterized";

**Manuscript was translated by professional interpreter.**

- p.12, l.12: "in particular";

**Manuscript was translated by professional interpreter.**

- p.13, l.8 "once a day": provide overpass time;

**It is not needed, because the satellite part was removed.**

- p.13, l.9 "scored specific place": rephrase. Line 10: "the square" -> "a square";

**Manuscript was translated by professional interpreter.**

- p.13, l.13: "is illustrated comparison of the annual averages..." -> "the comparison of the annual averages of total ozone is illustrated in Fig. 9". "it was the comparison of values obtained ... obtained": rephrase;

**Manuscript was translated by professional interpreter.**

- p.13, l.20: is 3.9 DU a "significant" difference? Line 21: "It can be said that...": please, explain better which sources of systematic errors have to be considered;

**It is not needed, because the ZS part was removed.**

- p.13, l.24: what value for the "Angstrom exponent" was used? Why?

**The explanation is in the manuscript (page 11, line 5).**

- p.13, l.27 "only AOD values smaller than 1.5": how was this threshold established?

**The explanation is in the manuscript (page 10, line 9).**

- p.13, l.28: I would say that a correlation index of 0.5 is moderate, not strong;

**It is not needed, because the satellite part was removed.**

- p.14, l.10: "was" -> "were". "so it is a period" -> "i.e. for a period".

**Manuscript was translated by professional interpreter.**

**Author's response to the comments of Anonymous Referee #2.**

**Authors comments are written in bold.**

1) General comments

The present paper includes a long UV AOD series spanning from 1994 to 2016 which may be useful to provide further insight on the role of aerosols on the Earth's climate. Furthermore, these data may be also used to demonstrate the capability of the Brewer spectrophotometer to measure AOD, most of these instruments being used only for ozone measurements.

In my opinion, these two points make the paper interesting for the scientific community. There are however three main issues:

a) There is still room to improve the scientific discussion, see points 2a-i below.

b) It is extremely important to show that the Brewer AOD is correct. For that, you need to provide meaningful comparisons with data from other instruments, see points 2j-k.

c) The quality of the presentation and, specially, of the English has to be improved, see Section 3 below.

Without improvements in these three areas, I can not support the publication of the paper in ACP. With a view to help the author improve the paper, I will provide specific questions and comments next.

2) Scientific discussion

a) On page 3, line 16, the author states that "In this case, the AOD calculation algorithm is part of the main control program for Brewer. The main difference from previous method is that the ETCs for individual wavelengths are not determined by LPM method but they are obtained during calibrating the instrument, i.e. every 2 years". Could the author explain how the ETCs are determined during this calibration of the instrument?

**This answer is included in the manuscript (page 7 , line 11): Their size is determined during calibration based on a comparison with the portable reference instrument No. 017.**

Have you compared the results from the LPM and the so-called calibration methods?

**Yes, I have. Please, see section 3.1 for details.**

b) On page 7, Fig. 1, there is an entry with the label "AOD-AAOD<0.5". This part of the screening algorithm does not seem to be explained in the text, and the definition of "AAOD" seems to be missing from the paper.

**Explanation of cloud screening is now included in the manuscript (page 10).**

c) On page 7, line 13, the author states that "Daily averages are calculated as arithmetic average of all values of a given day (from at least one value)." So, a daily average is considered valid even if there is just one AOD value for the day?

**Yes, a daily average is considered as valid.**

How many AOD measurements do you obtain on average for each day?

**It was obtained 7 AOD measurements for each day on long term average (1994 - 2016).**

How many times do you get just one measurement in a day?

**It was in 14% of days.**

d) From the discussion on page 8 about the calibration periods, it is not clear how they are selected. Are they the same as the period between the standard ozone calibrations?

**This answer is included in the manuscript (page 7 , line 14): The LPM applied in this paper employs fixed ETCs for a 2-year intercalibration period, which is identical with the standard intercalibration period for the measurement of ozone. It is assumed that any significant service modifications to the Brewer spectrophotometer during calibration may affect both the calculation of ozone and the calculation of AOD. For that reason, the period not exceeding 2 years was used.**

If so, does the author find that the stability of the ozone and AOD configurations are the same?

**The base assumption is that the stability of the ozone and AOD configurations are the same.**

Did the author try to use shorter calibration periods?

**No, I did not use shorter calibration periods.**

e) On Page 9, line 14, the author mentions that "ground measurements from the nearby station in Hradec Králové and satellite data" were used to complete the series down to 1962. What type of ground instrument operates at Hradec Králové? What satellite data was used?

**This answer is included in the manuscript (page 18 , line 13): It was possible to deduce the state of ozone as early as from 1962 by means of ground measurements from a nearby station in Hradec Králové (Dobson spectrophotometer, 1964–1978) and satellite data (Total Ozone Mapping Spectrometer, 1979–1993).**

Could the author show the full series from 1962 to 2016?

**It was added graphical illustration (page 18), please see Figure 9: Values of total column ozone amount (TCO) for Poprad-Gánovce, 1962–2016.**

f) From Figs. 2 and 4 and text, it's clear that the year 1994 is very noisy for the AOD data. Was this year included in the analyses (specially, the determination of the trends) in Section 3.2?

**Yes, this year was included in the analyses. Among other things, this answer is included in the manuscript (page 19 , line 15): It is obvious already at a glance that the AOD, unlike the total ozone optical depth, exhibits an apparent decline for the monitored period. For the wavelength of 306.3 nm, the value of the trend is –0.07 ±0.01 for 10 years, and the value of the trend for the wavelength of 320 nm is –0.06 ±0.01 for 10 years. If we omit first 2 years due to their lower reliability, then the value of the trend for the wavelength of 306.3 nm is –0.05 ±0.01 for 10 years, and the value of the trend for the wavelength of 320 nm is –0.04 ±0.01 for 10 years.**

g) Could the author provide some explanation for the behavior shown in Figs. 6 and 7? (E.g., are the peaks be related to weather conditions?)

**The behavior shown in figures was commented. For a details please see text near the figures.**

h) What conclusions does the author extract from Fig. 8?

**The behavior shown in figure was commented. For a details please see text near the figure.**

i) On page 13, line 20, the author states that "Such a significant difference is caused by inconsistent methodology for the calculation of total ozone through ZS measurements". Could the author elaborate further why does he consider the ZS measurement method inconsistent?

**The issue of ZS measurements has been removed from the manuscript. It was done because this issue is complicated and the scope of manuscript is moreover quite big.**

j) On page 13, line 27, while comparing the Brewer and OMI AOD data, the author writes that "The correlation coefficient has reached the value of 0.51 in comparison with each other what represents a strong positive correlation". I don't believe 0.51 should be considered a strong correlation. For comparison, what's the correlation coefficient between the Brewer and OMI ozone data? Instead of the bar plots in Fig. 9, could you plot the fits as in Fig. 10?

**The issue of comparisons with satellite measurements has been removed from the manuscript. It was done because this issue is complicated and the scope of manuscript is moreover quite big.**

k) As stated above, showing the readers that your Brewer AOD is correct is of the utmost importance. The comparison with the satellite data, as shown in this work and others before, sometimes might not be

straightforward. Making a comparison with other ground-based instruments would be thus a better option. According to the AERONET database, there is level 2.0 Cimel AOD data at 340 nm for the PopradGánovce site from December 2014 to January 2017, see https://aeronet.gsfc.nasa.gov/cgi-bin/type_one_station_opera_v2_new?site=PopradGanovce&nachal=0&year=22&aero_water=0&level=3&if_day=0&if_err=0&place_code=10&year_or_month=1

Could the author use these data to compare with the Brewer AOD? If not, could they provide a comparison with another ground-based instrument? If not, has the author considered the possibility of attending to some inter-comparison campaign?

**It was accepted. Please, see section 3.1 for details.**

3) Presentation

a) First and foremost, the quality of the English has to be improved. This is not a purely cosmetic question, there are sentences that are very difficult to understand, like e.g. "The lower limit of uncertainty was calculated using average value ETC from which its standard deviation has been deducted in the given calibration period" on page 9. Please, do check the whole paper and improve the English to an acceptable level.

**Manuscript was translated by professional interpreter.**

b) As mentioned before, it is critical to demonstrate that the Brewer AOD is correct. Such proof should come immediately after the AOD calculation method is presented and before any other discussion of the data. I thus suggest inserting Sec 3.3 (now including a comparison with AERONET data) before Sec 3.1

**It was accepted. Please, see section 3.1 for details.**

---

## Referee Report (RR1)

The manuscript underwent an impressive revision from both a stylistic and technical perspective.

As already said, the length of the dataset would normally deserve publication. However, I am concerned about the quality of some results, most notably:

1. the spectral dependency of the AOD cannot be reliably assessed, in my opinion, using a single-monochromator Brewer. For example, the method used by Arola and Koskela (2004) for roughly estimating the effect of the straylight on AOD at the shortest wavelengths, was employed by Peter Hrabčák to correct the measurements, which I think it is not in the intent of the authors of the original paper (the straylight primarily depends on the ozone slant path column, not on the solar zenith angle). Also, the author does not mention what are the conditions for which he calculated the correction for the light scattered within the instrumental field of view (forward scattering peak should be relevant for larger particles, but how large were the simulated aerosol particles?).

I think that the main outcome of the paper (reduction of the AOD and its contribution to the total optical depth) would not be compromised if the AOD for only the 320 nm wavelength, which is the most reliable measurement, were presented;

2. it is not very clear how the uncertainty of the annual averages (and thus, the resulting uncertainty/significance of the trend) is calculated. It is said that the error bars in Fig. 10 are obtained by changing the ETC within the range of its standard deviation. However, how is natural variability taken into account for the assessment of the significance of the trend? Moreover, using this method, the fewer ETC determinations for each year there are, the narrower is the uncertainty, which is probably not the desired behaviour of the method;

3. are the results of the standard lamp (SL) test used to recalculate ozone? This could maybe help in further improving the quality of the measurements, especially in periods when the Brewer is unstable.

From a stylistic point of view, I recommend a revision of the structure of the paper: Sect. 2.4 should be shortened (perhaps moving the most technical parts to the Appendix) and Sect. 3.3 should be anticipated before the results.

A complete list of the technical corrections will be provided once the final publication of the paper is foreseen.

---

## Editor Decision (ED1)

**Author's response to the comments of Anonymous Referee #1 and #2 and the track changes version.**

Author's response to the comments of Anonymous Referee #1. - bellow

Author's response to the comments of Anonymous Referee #2. - from page 11

The track changes version. - from page 14

**Author's response to the comments of Anonymous Referee #1.**

**Authors comments are written in bold.**

MAIN ISSUES

The paper cannot be published as is, and major revisions are required. I do not mean that the main outcomes of the work (trends of ozone and aerosol optical depths) are wrong, but several issues should be settled, or at least discussed, before the paper can be accepted.

15    1/ The determination of the ETC appears as the main problem, along with the estimate of its uncertainty:

1a/ the uncertainty of the mean, better taking into account the number of ETC estimates, should be used instead of the standard deviation of the ETC estimates. Otherwise, as said in the paper, the uncertainty can be lower for a time interval with few ETC estimates than for one with several estimates;

20    **The problem with a few, namely 2 ETC estimates was only in the first intercalibration period. This problem was solved by using additional data from the year 1993. Thanks to it the number of ETC estimates is sufficinnt now. The minimum number of ETC estimates for all intercalibration periods and all wavelnegths is 7 now. Therefore I think that the standard deviation of the ETC estimates is a good tool for the estimate of the uncertainty of AOD.**

25    1b/ in his reply to my review, the author states that "the values of the uncertainties of the annual averages do not enter the calculation of the uncertainty of the linear trend". Instead, I think that the uncertainties should enter the calculation of both the value and the uncertainty of the linear trend;

**Excuse me, but I do not know how to take correctly into account the uncertainties of the annual averages in the trend calculations.**

1c/ minimum data coverage thresholds (e.g., minimum measurement days per month, minimum measurement months per year) should be fixed before averaging the data (p. 11 l. 9-15), especially in case of large gaps (due to maintenance of the

instrument or clouds) and presence of a seasonal cycle. Maybe this is not an issue, but it should be specified how long is the longest period without data, if any;

**This sentences was added to the manuscript: „The longest period without data is 12 days for the ozone optical depth and 26 days for AOD. There was no month without data. In both cases, February 2013 was the month with the absolute lowest number of days with the determined optical depth. In the first case, it was 10 days, and it was 3 days in the second case. With regard to the long-term average, December is the month with the lowest number of days with the determined AOD with 13 days. The opposite extreme falls on August with 24 days."**

1d/ I could have missed this information, but I do not understand why "an advantage of the LPM is that it takes into account the change of ETCs during the intercalibration period" (p. 16 l. 19-20). From Fig. 4, I can understand that the ETCs estimated by the LPM are averaged and fixed for an intercomparison period, aren't they?

**Maybe the problem was my bad formulation and therefore I have changed related text in the manuscript a little bit: „An advantage of the presented LPM is that it partially takes into account the potential change in the sensitivity of the instrument during the intercalibration period, because ETCs are determined based on real measurements in a given period. The BSM utilizes ETCs determined during calibration. Therefore, the potential change in the sensitivity of the instrument during the intercalibration period is not taken into account at all"**

1e/ is the Langley plot performed by correcting S_lambda (Eq. 3) for O3 absorption or by including O3 in tau_lambda? In the first case, why the ozone cycle is mentioned as a source of error (p. 5)? In the second case, how is mu_w calculated (including ozone)?

**O3 is including in $\tau_\lambda$. This explanation is included in the manuscript: „$\mu_w$ is the air mass factor for atmosphere as a whole. Its value was calculated as a weighted arithmetic average of individual aforementioned components, while the optical depth of a given component was the weighting factor." Moreover, the text located after equation (4) was enriched about an information related to calculation of $\mu_w$: The final values of ETCs used to calculate the AOD were available only after two iterations. At the beginning, initial values of AOD were not available yet. For this reason, it was not possible to apply the second and third criterion and it was also not possible to take into account the effect of aerosols in the calculation of $\mu_w$ and to determine the correction for the diffuse radiation. It was possible to do so on the first and second iteration.**

1f/ since the BSM method is basically a calibration transfer from a reference instrument, the author should explain how Brewer #017 was calibrated;

**This explanation was added to the manuscript: " ETCs for the reference instrument are determined by LPM at the Mauna Loa or Izaña observatory."**

2/ the AOD differences between the LPM and the BSM methods can be easily attributed to the ETC difference (AOD difference = 1/mu * log(ETC1/ETC2)). Therefore, the deviations between the two methods are mainly not "random" (contrary to what is written at p. 15 l. 14 and at p. 16 l. 12); the observed variability between summer and winter (p. 15 l. 21-23) is an obvious consequence of air mass variation during the year; the AOD differences, always positive in the first period and always negative in the second period, are dictated by the difference in ETC shown in Fig. 5. I would thus recommend to plot the AOD differences as a function of air mass to check that the ETC difference is the main cause of deviation;

**Note in the bracket at p. 15 l. 14: „so-called large random error" was removed. Word "random" was removed in related sentence (p. 16 l. 12) and the sentence was reword. I have investigated the AOD differences as a function of air mass for the longest wavelength. Possible causes are discussed in the manuscript, included ETCs difference. It has been found that: " The values of the determined ETCs affect the observed differences the most. However, it is true only if their difference between LPM and BSM is sufficiently large." It was added to the manuscript.**

3/ as I mentioned in my previous review, an additional source of errors can be the missing SL correction in the BSM, which is not mentioned among the sources listed at p. 16. However, according to the maintainer of the Brewer operating software: "Using the real-time AOD (or even ozone) data from the operating software for anything other than a rough idea about the AOD doesn't make any sense since no corrections are applied in real time for the SL ratio changes. It's likely worse than real time data for ozone due to potential changes in absolute sensitivity in Brewers (that do not affect ozone)" (V. Savastiouk, personal communication, 2017). This issue should be discussed and its potential impact estimated;

Using the real-time AOD (or even ozone) data from the operating software for anything other than a rough idea about the AOD doesn't make any sense since no corrections are applied in real time for the SL ratio changes. **My comment to this sentence: I know that BSM has a lot of disadvantages, but it was used also for calculation of AOD in the study Kumharn et al., 2012.**

It's likely worse than real time data for ozone due to potential changes in absolute sensitivity in Brewers (that do not affect ozone). **My comment to this sentence: I know (I asked IOS) that SL correction can by used for ozone calculation but no for for AOD calculation (for counts rates corrections) because standard lamp has been losing its intensity over time. For example, we observe an decrease in count rates, but we don't know if the reason is a change in absolute sensitivity in Brewer or a change in the intensity of standard lamp.**

**Deep analysis of the influence of missing SL correction in case of BSM is outside the scope of this paper. But it was mentioned as a potential source of error. This sentence was added to the manuscript: " The disadvantage of BSM is also the absence of the SL (standard lamp) test that leads to inaccurate TCO values, which consequently affects the size of AOD."**

4/ even though it is probably a minor issue (especially at low air masses), the spectral stray light effect is inaccurately taken into account, in my opinion, for two reasons:

4a/ both the slits employed for UV scans and the grating positions are different from those of DS measurements, therefore the ratio between the irradiance at longer and shorter wavelengths in the "scanning mode" may not be representative of stray light in "slitmask" mode (DS measurements). The cited study of Arola & Koskela (2004), using this method, only aimed at a rough estimate of the effect;

**This sentence was added to the manuscript: " The potential impact of a different measurement mode was neglected for the sake of simplification."**

4b/ the assumption of spectrally constant stray light is simplistic, as pointed out in Garane et al. (2006), and will lead to some uncertainty.

**The end of part "2.4.1 Correction for the diffuse radiation and stray-light effect" was changed: " In view of simplification, it is also assumed that the value of stray light is constant for all wavelengths. The said simplifications may lead to some uncertainties when estimating the stray light effect. These uncertainties were not investigated further."**

Therefore, if the author wants to keep the analysis of wavelengths shorter than 320 nm, they should point out that they could be affected by the stray light, rather than trying to correct them and jump to the conclusion that the Angstrom coefficient is negative;

**I have decided to let the stray light correction in the manuscript. Now, I know that is not perfect, but I think that it is better like no correction. In the Conclusions is stated that the Ångström exponent is probably negative mainly due to the size and accuracy of ETC determination: "It was also determined that the application of corrections to the diffuse radiation, stray-light effect and polarization decreased the difference between the value of AOD for the shortest and longest of the examined wavelengths. Although the difference has been reduced, it still reaches negative values, which results in the negative values of the Ångström exponent as well. With regard to the long-term average for 23 years, the difference in AOD is only –0.006. The application of the correction for the polarization had the most significant influence on the reduction of the given difference. On the contrary, the lowest contribution was achieved by the correction with regard to the diffuse radiation. The key factor influencing the value of the examined difference is probably the size and accuracy of ETC determination."**

5/ most of all, language needs major revisions, as well as the structure of the paper:

5a/ check the whole text for the correct use of articles "a" and "the";

**Several sections of the text have been repaired.**

5b/ remove all technical details that are not fundamental for the paper (e.g., discussion about wavelengths, p. 4) and cite some reference where the information can be found elsewhere (e.g., instead of the data reduction explained at p. 8);

**Some parts (not only technical) of the manuscript has been removed, namely:**

5    **Page 2: „The direct impact means scattering and absorption of shortwave and longwave radiation. Absorption of radiation subsequently leads to warming of those atmospheric parts, where aerosols are present (primarily in the boundary layer of the atmosphere) and higher temperature consequently leads to evaporation of cloud layers. The last sentence concisely described the semidirect impact, resulting in a higher density of solar radiation flow reaching the Earth's surface (Cazorla et al., 2009). The higher temperature may lead to a change of thermal stratification of**
10   **the atmosphere, which consequently affects vertical and horizontal movements of air in the atmosphere. The indirect impact pertains to the ability of aerosols to act as condensation nuclei or ice nuclei, which affects microphysical and optical properties of clouds. Yet, it concerns a change of their radiation characteristics, change of atmospheric precipitation characteristics, and alteration in cloud lifetime as well. An increase in the number of condensation nuclei leads to the rise of cloud droplet count and to the reduction of their size under the given conditions of water**
15   **content in the atmosphere, causing an increase of albedo and extension of cloud lifetime (Lohmann and Feichter, 2005; Unger et al., 2009)."**

**Page 4: „The spectral separation of solar radiation is carried out by means of a modified Ebert f/6 spectrometer, which uses a holographic diffraction grating with a resolution of 1,200 lines per one mm (Sci-Tec, 1999). The instrument stability of Brewer spectrophotometer measurements is ±0.01 nm, namely throughout the temperature**
20   **range (Sci-Tec, 1999). The value of the smallest wavelength increment (microstep) is 0.006 ±0.002 nm (Sci-Tec, 1999)."**

**This sentence has been added instead of the removed text : „More informations about the instrument and the measurement process can be found in Kerr (2010)."**

**Page 4: „During the monitored 23 years, the sizes of wavelengths were refined five times in total. Their accurate sizes**
25   **were used at the beginning of the examined period. In 1994, the sizes were 306.276 nm, 310.04 nm, 313.494 nm, 316.799 nm and 319.999 nm. The end of the period, that is 2015, was characterized by values of 306.276 nm, 310.035 nm, 313.476 nm, 316.755 nm and 319.989 nm. Observed deviations are only minimal. Therefore, the aforementioned long-term average is generally applicable to the whole period."**

30   5c/ split the theory ("Methods") from the results. E.g., p. 11 l. 19-26 are theory, as well as p. 12 l. 5-14;

**It was accepted. This new part of subchapter 2.4 has been created: "2.4.1 Correction for the diffuse radiation and stray-light effect". Moreover, the main structure of subchapter 2.4 has been changed. It means that, also other parts of subchapter 2.4 has been created, namely:**

**2.4.2 Calculation of extraterrestrial constants**

**2.4.3 Quality control process**

**2.4.4 Additional notes**

5d/ use letters, e.g. (a), (b), etc., not "right/left", to identify the subfigures in the panels, according to the

ACP guidelines

(https://www.atmospheric-chemistry-and-physics.net/for_authors/manuscript_preparation.html).

**It was accepted.**

TECHNICAL CORRECTIONS

- replace "Operational Brewer Program" with "Brewer Operating Software";

**It was accepted**.

- better use "AERONET" instead of "Cimel", since the data are provided by AERONET;

**I did not accept it because "Cimel sunphotometer" is common in another scientific publications, for example:**

**De Bock, V., De Backer, H., Mangold, A., and Delcloo, A.: Aerosol Optical Depth measurements at 340 nm with a Brewer spectrophotometer and comparison with Cimel sunphotometer observations at Uccle, Belgium, Atmos. Meas. Tech., 3, 1577-1588, https://doi.org/10.5194/amt-3-1577-2010, 2010.**

**López-Solano, J., Redondas, A., Carlund, T., Rodriguez-Franco, J. J., Diémoz, H., León-Luis, S. F., Hernández-Cruz, B., Guirado-Fuentes, C., Kouremeti, N., Gröbner, J., Kazadzis, S., Carreño, V., Berjón, A., Santana-Díaz, D., Rodríguez-Valido, M., De Bock, V., Moreta, J. R., Rimmer, J., Boulkelia, L., Jepsen, N., Eriksen, P., Bais, A. F., Shirotov, V., Vilaplana, J. M., Wilson, K. M., and Karppinen, T.: Aerosol optical depth in the European Brewer Network, Atmos. Chem. Phys. Discuss., https://doi.org/10.5194/acp-2017-1003, in review, 2017.**

- p. 1: use the long form "spectral stray light" instead of just "stray light" at the first occurrence, to differentiate it from other sources of stray light;

**It was accepted.**

- p. 3: latitude °N and longitude °E (NW/EW?);

**It was accepted.**

- p. 3: order the pollution sources by relevance;

**The local pollution sources are mentioned by relevance. It means solid fuel combustion and the agriculture as first and second. Industrial activities in near city as third.**

- p. 4: unless very old instruments are considered, UV, SO2 and O3 estimates are performed by all instruments. NO2 instruments are performed only by MKIV Brewers, which are no longer manufactured;

**The problem sentence has been corrected: The model MKIV allow moreover measuring the vertical column of SO$_2$, NO$_2$ (for this purpose, measurements of solar radiation are made in the visible region of the spectrum) and global UV radiation as well (from 290 to 325 nm, with an increment of 0.5 nm).**

- p. 5: it is said that LPM is not recommended for low latitude stations, however Poprad-Gánovce (706 m a.s.l.) cannot be considered a high altitude station. Please, clarify this point;

**I agree, that the Poprad-Gánovce station (706 m a.s.l.) can not by considered as a high altitude station but also is not a typical low altitude stations in urbanized area.**

**Related text in the manuscript was enriched. This description in the manuscript clarify your remark: „In the light of the listed errors, it is not recommended to determine extraterrestrial constants (ETCs) using the Langley plot method (LPM) for low altitude stations in urbanized areas, unless corrections for the impact of diffuse radiation and the daily cycle of ozone are known. Poprad-Gánovce is not a typical station in a low-altitude urbanized area and is not even a typical high mountain station. It is between these two cases. The impact of daily ozone cycle and NO$_2$ can be neglected considering the rural location and higher altitude of the Poprad-Gánovce station. The diffuse radiation impact and stray-light effect were not neglected."**

- p. 5: how were the air mass thresholds of 3 (line 12) and 4 (line 32) established?

**The scientific reference was added to the manuscript: „A recommendation to ensure the air mass factor does not exceed the value of 3 in the calculation of ETCs was taken into account as well (Arola and Koskela, 2004; De Bock et al., 2010)."**

**Also the change related to threshold 4 has occurred in the manuscript: „A DS measurement is accepted only with the relative optical mass of less than 4 (as recommended by IOS) and it takes approximately 2.5 minutes."**

- p. 5: explain better the "polarization effect" and use the form "internal polarization effect";

**The related part of the manuscript was enriched about this explanation: „It occurs due to the combined effect of two polarization sensitive elements of the Brewer spectrophotometer. The entrance window is the first of mentioned elements and the grating is the second. Polarization effect depends on the zenith angle of the Sun and is almost wavelength independent."**

**The form "internal polarization effect" has used at the first occurrence instead of the form "polarization effect".**

- p. 5: "corrections published in Cede at al. (2006)" -> which one of the methods described in the paper was used?

The name of the used method was added in to the brackets. „As a result, corrections (the field measurements method) published in Cede et al. (2006) could be applied to the instrument No. 97.“

- p. 6: include the physical units for the Loschmidt constant;

It was accepted: „$n$ is the molecule count in the volume determined by 1 DU and 1 cm$^2$; for O₃, it is a constant with the value of n $= 2.687 * 10^{16}$ cm$^{-3}$“

- p. 7: remove repetitions in lines 1-3;

Some changes has occurred in the manuscript. But I'm not sure if I understand your request properly (for more information about the changes in the text, please see page 6, lines 22-27 ).

- p. 7: explain why the method works best at lower latitudes;

The reasons are mentioned in the previous text: „It is also necessary to avoid an impact of cloudiness on direct solar radiation (DS) measurements and to ensure a sufficient scope of zenith angles of individual DS measurements during the day, which is needed for the given method.

For the said reasons, this method is most appropriate for lower latitudes (especially in mountainous regions near the tropics), and it has certain limitations in the middle and particularly higher latitudes (Nieke et al. 1999; Marenco 2007).“

- p. 7: "implemented", not "developed" by Vladimir Savastiouk;

It was accepted.

- p. 7: are the alternative Komhyr (1980) and Kasten and Young (1989) formulations used for O3 as well?

This explanation was added to the manuscript: „Brewer Operating Software and O3Brewer software use the same formula to calculate an airmass factor for ozone. The same applies to Rayleigh scattering. The given formulas are defined in the Brewer MKIV Spectrophotometer Operator's Manual (Sci-Tec, 1999). Different, yet on the other hand more accurate formulas, were used to calculate the AOD by LPM – for ozone according to Komhyr (1980) and for Rayleigh scattering according to Kasten and Young (1989).“

- p. 8: why was the stray light correction applied at that stage?

This explanation was added to the manuscript: „After the deadtime compensation, a correction was applied to the stray-light effect, in the same order as Garane et al. (2006).“

- p. 8: explain if ND filter spectral transmittance was taken into account for AOD calculations;

**A change has occurred in the manuscript. Related text including explanation is: „In the fourth step, a correction for the temperature dependence was applied, including a spectral transmittance correction for the utilized neutral density (ND) filter. These filters are automatically selected by the Brewer spectrophotometer with respect to the current density of the solar radiation flow. There are 5 ND filters and 5 wavelengths as well, so 25 attenuation values**

5    **are needed in total. The attenuation values of given filters are determined during calibration of the instrument.“**

- p. 8: what are the "practical purposes"?

**The sentence has been changed: „Theoretically, it is feasible to determine the unknowns already from two measurements, but for the ETC determination, it is advisable to acquire as many measurements as possible.“**

- p. 9: the conditions 1-3 are related to the single measurement, while conditions 4-9 concern the whole day. They should be splitted;

**It was accepted. The conditions has been splitted.**

15    - p. 11: "more relevant" compared to what?

**The sentence was enriched about the explanation: „The main goal of this calculation was to acquire a comparison of AOD for the same wavelength, which is more relevant than a direct comparison of AOD for two different wavelengths.“**

20    - p. 11: include a formula for the correction factor;

**I think that a formula is unnecessary, because the correction factor is good explained in the text (one sentences was added): „A ratio of the circumsolar radiation to the direct solar radiation was calculated using the SMARTS 2.9.5 programme (available at https://www.nrel.gov/rredc/smarts/). The calculations in SMARTS were implemented for rural aerosol conditions, which are characterized by the Ångström exponent equal to 0.96. The ratio of the**

25    **circumsolar radiation to the direct solar radiation was determined for all five wavelengths. This ratio is hereinafter referred to as the correction factor. The values of zenith angle, aerosol optical depth and atmospheric air pressure were taken into account in the calculation of the correction factor. It follows that this factor was determined for specific conditions at a given time.“**

30    - p. 14: what criterium was followed to define "similar" the two ln(ETC)'s, sometimes deviating by 0.1 (l. 19)?

**Related sentence was reworded: " It can be seen that ETC values show some differences."**

Similarly, how can you quantitatively tell that the agreement is "excellent" (p. 15)

**Related sentence was reworded: „A good agreement of both methods was seen in the first period.“**

and "high level"?

**Related sentence was reworded: „On the other hand, good agreement is proven by a median and standard deviation (1σ) of the differences in AOD values for the LPM and BSM with the values of –0.004 and 0.01."**

Compare your results to the ones obtained in a recent paper about Brewer AOD in the UV range (of López-Solano et al., 2017, https://www.atmos-chem-phys-discuss.net/acp-2017-1003/);

5    **As you recommended, I have inspired by the article López-Solano et al., 2017. Instead of average of differences now I use the median of differences together with standard deviation (1σ) of differences.**

- p. 16: "it is not known that the BSM would take into account a change in the distance of Earth from the Sun". Does it mean that the Brewer Operating Software does not take into account the Earth-Sun distance or that you do not know whether it

10    takes this factor into account? In the latter case, examine the source code or ask to the maintainer of the Brewer operating software!

**Now I have confirmation from IOS (International Ozone Services) and therefore I have changed the sentence: „The first cause is that the BSM does not take into account a change in the distance of Earth from the Sun."**

15    - p. 18: since only direct sun measurements are considered in the paper, only report the ozone trend for DS;

**It was accepted. The ozone trend and also 5-years averages was corrected only for DS measurements. Related text in the manuscript has been changed: „If only DS measurements are taken into consideration, the average value of total ozone was 326 DU in the last five years (2012–2016), which is only 1 DU more than in the first five years of the monitored period (1994–1998). The linear trend for the period of 1994–2016 has the value of 0.8 ±2.2 DU for 10**

20    **years."**

- p. 21: explain what agricultural activities can determine the increase of AOD (wildfires?).

**Related text in the manuscript has been changed and enriched by an explanation: „Higher values of AOD in the months of April to September are presumably related to agricultural activities in the vicinity of the station. The**

25    **location is windy and a bare dry soil or even plant products are often blown away."**

**Author's response to the comments of Anonymous Referee #2.**

**Authors comments are written in bold.**

This new version of the paper shows many improvements in its organization and readability. I have some remaining questions and suggestions, but they are fairly minor. Once they have been taken into consideration by the author, I believe the paper will be suitable for publication.

1) General questions:

a) Could you insert in the manuscript a brief explanation for the approximation of the aerosol airmass presented on page 7, line 30 (that is, $\mu_a \sim \mu_{H_2O}$)?

**A brief explanation was added to the manuscript: „$\mu_a$ is the air mass factor of aerosols ($\mu_a < 4$ for AOD), while due to the similar vertical profile of aerosols and water vapor it holds that $\mu_a \cong \mu_{H_2O}$, where $\mu_{H_2O}$ is the air mass factor of water vapor determined according to Kasten (1966).“**

b) Could you provide some justification (a reference would suffice) for the sentence on page 8, line 2, stating that the $SO_2$ has a low impact on the AOD determination and its determination is not accurate?

**A reference was added and also a little change was made in the sentence: „The contribution of sulfur dioxide was neglected, namely due to its low impact (Arola and Koskela, 2004) and due to its inaccurate determination at the Poprad-Gánovce station as well.“**

c) Why do you call the algorithm presented in page 10 a "cloud screening"? It clearly removes a lot of bad measurements, but are these measurements only produced by clouds?

**The algorithm presented in the Figure 1 was renamed to quality control process. Related text was therefore corrected.**

d) On page 16, line 14, it is stated that "it is not known if the BSM would take into account a change in the distance of Earth from the Sun". Have you asked IOS?

**Now I have confirmation from IOS and therefore I have changed the sentence: „The first cause is that the BSM does not take into account a change in the distance of Earth from the Sun.“**

2) Figures:

a) On Fig.4, the line used for the average ETC shows some sloped segments. However, the average ETC are constant values in every two-year period. The sloped segments should be removed, leaving a blank space if necessary. This would also help to identify the calibration periods.

5 **It was accepted.**

b) Figs. 6, 7, and 8: I suggest adding the date range above on top of the figure for easier reference.

**It was accepted.**

10 c) Figs. 7 and 8: the label on the vertical axes should be "AOD – CSP" and not "AOD – fotometer"

**It was accepted.**

3) English: please check the following

a) Page 1, line 7: instead of "presented", it should be "present"

15 **It was accepted**

b) P.1, l.10: remove the blank space between "MK" and "IV", as in the rest of the paper

**It was accepted**

c) P.7, l.6: I think "mountainous" is more usual than "montane"

**It was accepted**

20 d) P.7, l.10: remove "a" from "is a part"

**It was accepted**

e) P.7, l.13: change "size" to "value"

**It was accepted**

f) P.7, l.14: change "reference instrument" to "reference Brewer instrument"

25 **It was accepted**

g) P.8, l.5: change "An NOAA" to "A NOAA"

**It was accepted**

h) P.8, l.23: change "Eq. (2) is adjusted" to "Eq. (2) is linearized"

**It was accepted**

30 i) P.8, l.31: change "The inclination of obtained line a" to "The slope a of the obtained line"

**It was accepted**

j) P.8, l.32: change "The natural logarithm of ETC ln(S0,\lambda)" to "The natural logarithm of the ETC, ln(S0,\lambda)"

**It was accepted**

k) P.11, l.19: change "was made pursuant to the recommendations" to "was made following the recommendations"

**It was accepted**

l) P.14, l.5: change "as well ETCs" to "as well as the ETCs"

**It was accepted**

m) P.18, l.15: change "by means of an average value of ETC" to "by means of the LPM method using an average ETC value for every 2-year period shown in Fig.4"

**It was accepted**

n) P.18, l.17: change "by analogy" to "in the same way but adding the value of the standard deviation"

**It was accepted**

o) P.20, l.5: change "In case of AOD" to "In the case of the AOD"

**It was accepted**

p) P.21, l.4: change "Figure 13 presents the same characteristics as in Fig. 12" to "Fig. 13 is the same as Fig. 12"

**It was accepted**

q) P.21, l.17: remove the commas enclosing "for which it was possible to determine the AOD"

**It was accepted**

r) P.22, l.21: change "Cloud screening employed" to "The cloud screening method employed"

**It was accepted**

s) P.23, l.5: remove "of the two compared methods"

**It was accepted**

t) P.23, l.7: change "longest of examined wavelengths" to "longest of the examined wavelengths"

**It was accepted**

u) P.23, l.9: change "application of correction" to "application of the correction"

**It was accepted**

v) P.23, l.14: change "The slightly" to "A slightly"

**It was accepted**

[revised manuscript text omitted]

---

## Author Response (AR2)

**Author's response to the comments of Anonymous Referee #1 and #2.**

Author's response to the comments of Anonymous Referee #1. - bellow

Author's response to the comments of Anonymous Referee #2. - from page 4

**Author's response to the comments of Anonymous Referee #1.**

**Authors comments are written in bold.**

The manuscript underwent an impressive revision from both a stylistic and technical perspective.

As already said, the length of the dataset would normally deserve publication. However, I am concerned about the quality of some results, most notably:

1. the spectral dependency of the AOD cannot be reliably assessed, in my opinion, using a singlemonochromator Brewer. For example, the method used by Arola and Koskela (2004) for roughly estimating the effect of the straylight on AOD at the shortest wavelengths, was employed by Peter Hrabčák to correct the measurements, which I think it is not in the intent of the authors of the original paper (the straylight primarily depends on the ozone slant path column, not on the solar zenith angle).

**I have followed an instructions in Arola and Koskela, 2004 and Garane et al., 2006. Stray-light effect corrections was calculated according to this inscructions. I did not find in this articles, that the stray-light effect primarily depends on the ozone slant path column and not on the solar zenith angle.**

Also, the author does not mention what are the conditions for which he calculated the correction for the light scattered within the instrumental field of view (forward scattering peak should be relevant for larger particles, but how large were the simulated aerosol particles?).

**Correction for the diffuse radiation was calculated using the SMARTS 2.9.5 programme (available at: https://www.nrel.gov/rredc/smarts/). The calculations in the SMARTS were implemented for the rural aerosol conditions, which are characterized by the Ångström exponent equal to 0.96 (this information was added to the munascript: page 11, line 22). This Ångström exponent value characterizes the size of the aerosol particles.**

I think that the main outcome of the paper (reduction of the AOD and its contribution to the total optical depth) would not be compromised if the AOD for only the 320 nm wavelength, which is the most reliable measurement, were presented;

**Thank you for your recommendation. I have decided, that I do not accept it. The main reason is, that it would lead to considerable depletion of the total content of the manuscript.**

2. it is not very clear how the uncertainty of the annual averages (and thus, the resulting uncertainty/significance of the trend) is calculated. It is said that the error bars in Fig. 10 are obtained by changing the ETC within the range of its standard deviation.

**Yes, it was a way how the uncertainty of the annual averages was calculated. The error bars was obtained by changing the ETC within the range of its standard deviation. The lower limit of uncertainty was calculated by means of an average value of ETC, from which its standard deviation for the given intercalibration period was deducted. The upper limit of uncertainty was determined by analogy. A linear trend was calculated by means of a linear regression using the least squares method. The significance of the trend is defined by the uncertainty of the linear trend. The uncertainty of the linear trend is defined by standard deviation ($\pm\sigma$) of the inclination of obtained line (a little change was occured in the manuscript: page 11, line 14). The values of the uncertainties of the annual averages do not enter the calculation of the uncertainty of the linear trend.**

However, how is natural variability taken into account for the assessment of the significance of the trend?

**The natural variability was eliminated by annual averages. The annual averages were calculated as an arithmetic average of individual monthly values.**

Moreover, using this method, the fewer ETC determinations for each year there are, the narrower is the uncertainty, which is probably not the desired behaviour of the method;

**The range of uncertainty interval depends primarily on suitable weather conditions in the given intercalibration period, as well as on the stability and homogeneity of measurements on days, when it was possible to determine the ETC. The number of days, when it was possible to determine the ETC, plays its role too. For instance, there were only two measurements in the first intercalibration period (it covers 1994 and a smaller part of 1995), and for that reason, inter alia, the uncertainty interval for 1994 is narrow and has a very low relevance. In other intercalibration periods, there were at least 8 ETCs. Therefore, the following reliability intervals can be deemed trustworthy (this part is in the manuscript: page 18, line 17).**

3. are the results of the standard lamp (SL) test used to recalculate ozone? This could maybe help in further improving the quality of the measurements, especially in periods when the Brewer is unstable.

**The results of the standard lamp (SL) test has used to recalculate ozone. The total ozone was calculated using the Brewer Spectrophotometer B Data Files Analysis Program software v. 5.0 by Martin Stanek (http://www.o3soft.eu/o3brewer.html). This software allows SL correction.**

From a stylistic point of view, I recommend a revision of the structure of the paper: Sect. 2.4 should be shortened (perhaps moving the most technical parts to the Appendix)

**The sect. 2.4 was shortened by removed some equations and its explanations. The reader can find this equations and its explanations in the attached references. The structure of the sect. 2.4 was also revised due to above mentioned changes.**

and Sect. 3.3 should be anticipated before the results.

**I have also took into account the recommendation of the Anonymous Referee #2: "In short, I suggest to reorder the sections in the following way: 3.3 (corrections), 3.2 (ETCs), 3.1 (validation), 3.4 (long AOD series)". Therefore the sect. 3.3 is now the first part of the Results and discussion. I think that it is better to let sect. 3.3 in the beginning of the Results and discussion, how it is recommended by the Anonymous Referee #2.**

A complete list of the technical corrections will be provided once the final publication of the paper

is foreseen.

**Author's response to the comments of Anonymous Referee #2.**

**Authors comments are written in bold.**

With respect to the previous version, the paper has improved significantly. The author has followed the suggestions of the referees and provided reasonable answers to most of their questions. There are, however, points that still need further attention, as discussed below.

In my opinion, the main outstanding issues of the paper are:

1) On Sec. 3.1, the author presents a comparison between the AOD determined by three methods - his own, the one implemented in the standard Brewer operating software, and a Cimel photometer. This is a great addition which provides the necessary confirmation of the quality of the present data. I have, however, some questions and suggestions with regard to this section:

i) First, I suggest moving this section to just before Sect. 3.4. The current sections 3.2 and 3.3 provide information on the calibration and corrections used to determine the AOD, and section 3.4 provides data retrieved using them, so it should come after them, and just before the whole series is shown. This change would also make easier to understand Sect. 3.1, because the reader would have been introduced to the details of the ETCs determination which are currently provided in Sect. 3.2. Furthermore, I would also suggest changing the order of Sects. 3.2 and 3.3, because it seems that the determination of the ETCs in Sect. 3.2 requires the data corrections explained in Sect. 3.3.

In short, I suggest to reorder the sections in the following way: 3.3 (corrections), 3.2 (ETCs), 3.1 (validation), 3.4 (long AOD series)

**It was accepted.**

ii) For the 2015-2016 period, the BSM-LPM fit on Fig. 2 doesn't seem to show any offset, the CSP-LMP fit on Fig. 3 shows a small one, and the CSP-BSM fit on Fig. 4 shows a large one. This seems strange at first sight. Can the author provide an explanation?

**These average differences are the primary reason for observed offsets on attached charts. As a result of the offset, the intersection of the linear fit is not the same as the intersection of the main axes of the graph. This is the best illustrated by the right graph in Fig. 8, because in this case the average difference has the highest absolute value of all presented comparisons (the same explanation was added to the munascript: page 17, line 15).**

What are the values of the intercepts (independent terms) of the fits?

**The value of the intercept of the fit is 0.025 for the CSP-LMP comparison (a small offset) and the value of the intercept of the fit is 0.074 for the CSP-BSM comparison (a large offset). The equations of the fits was added to the graphs (Figure 6, 7 and 8). This equations contains the values of the intercepts.**

2) The method used to calculate the stray light correction in pages 17-18 does not seem clear to me. In particular, on page 17 the author states that "A ratio of average count rates for four wavelengths in the region from 290 to 291.5 nm to the count rates for the monitored wavelength (one out of five) was determined for 3,386 spectral analyses in total, as well as for various zenith angles of the Sun within them". The author always refer to 306-320 nm as the Brewer operational range, so what are these four wavelengths from 290 to 291.5 nm?

**Brewer ozone spectrophotometer (MK IV) performs standard measurements of direct solar radiation (DS) in the UV region at five selected wavelengths, namely 306.3 nm, 310 nm, 313.5 nm, 316.8 nm and 320 nm. It measures also global UV radiation from 290 nm to 325 nm, with a step 0.5 nm (this information was added to the munascript: page 4, line 7 ).**

Why does the calculated ratio provide the stray light correction?

**I have added some important notes to the manuscript (for more information please see page 12, line 5). It was not possible to explain all detail because the extent of the munascript is big, but you can find more informations in the article by Arola and Koskela, 2004.**

3) Figure 9 on page 18 provides a long TOC series from 1962 to 2016, made up of data from a Dobson spectrophotometer operating from 1964 to 1978 at a nearby site, satellite data from TOMS for the 1979-1993 period, and Brewer data for the years 1994 to 2016. Although I requested this figure in my previous review, now I have some concerns regarding it, in part because of the changes introduced in this revision of the manuscript:

i) If the period of operation of the Dobson is 1964-1978 as written on page 18, where does the data for the years 1962 and 1963 come from?

**Excuse me, it was a mistake. Correct period for Dobson is 1962-1978.**

ii) Has the author checked that all the three datasets used in the figure are compatible? The Dobson and the Brewer instruments operated at different sites, and ground-based and satellite data can, in my experience, present substantial differences.

**You are right, there is also a problem with the compatibility of the three datasets. The compatibility is not clear.**

iii) Last but not least, what conclusions should the reader extract from Fig. 9? The focus of this new version of the paper seems to be the determination and analysis of a long Brewer AOD series at Poprad-Gánovce. This figure only seems to allow the author to write that the year 1994 is the minimum of the combined series, but this is

doubtful because this year is the least reliable one of the Brewer AOD data, as discussed on page 19. I would thus now suggest removing this figure and related text to keep the focus on the main message of this revision of the paper - the Brewer AOD calibration and resulting data series.

**Figure with TOC series from 1962 to 2016 was removed. Also removed was the text related with TOC series from 1962 to 2016. In the text has been made the another changes too (for more information about the changes in the text, please see page 18, line 6).**

Besides the main issues mentioned above, there are some smaller ones that should be also addressed:

4) The quality of the English has been greatly improved, but there are still some problems, like e.g. a missing "a" in front of "Cimel" on line 14 of the abstract, a missing "the" in front of "five" in the line 31 of page 5, or a missing "the" in front of "more" on line 7 of page 14, to mention a few. I recommend the author to check again the whole text.

**The mentioned mistakes were corrected and the whole text was checked again.**

5) In Eq. 2, two different symbols, "m" and the greek letter "\mu", are used to denote the airmasses of different contributions. Although the symbols are explained in the text, it would be easier for the reader to keep just one of them and use subindices to differentiate among the contributions.

**It was accepted. For more information please see Eq. 2 (page 7) and related text.**

6) The equation of point 8 at the top of page 9 should be explained in words. Also, why was the value 1.75 selected?

**Related text was rewrote and enriched about some explanations. Threshold value 1.75 was also explained (for more information please see page 9, line 12).**

7) Eqs. 5-7 seem to correspond to the airmasses introduced in Eq. 2. Why "AMF" is used now instead of "\mu"?

**AMF in the equations was replaced by the greek letter "\mu".**

It would also be better to introduce these definitions just after Eq. 2.

**Now the content related to airmass factors follows very close behind the equation 2. But, I have also took into account the recommendation of the Anonymous Referee #1 about the truncation of content of the sect. 2.4. Therefore the sect. 2.4 was shortened by removed this equations and its explanations. The reader can find this equations and its explanations in the attached references. The structure of the sect. 2.4 was also revised due to above mentioned changes.**

8) On page 21, line 6, the author states that "the observed characteristics are typical for the central European location of the station". Why? Could the author explain this in more detail and provide some reference to back up this statement?

**Related text was rewrote and enriched about a reference (for more information please see page 20, line 18).**